# VAEM: a Deep Generative Model for Heterogeneous Mixed Type Data

**Chao Ma** *
University of Cambridge
Cambridge, UK
cm905@cam.ac.uk

**Sebastian Tschiatschek** *
University of Vienna
Vienna, Austria
sebastian@tschiatschek.net

**Richard Turner** *
University of Cambridge
Cambridge, UK
ret26@cam.ac.uk

**José Miguel Hernández-Lobato** *
University of Cambridge
Cambridge, UK
jmh233@cam.ac.uk

**Cheng Zhang**
Microsoft Research Cambridge
Cambridge, UK
cheng.zhang@microsoft.com

## Abstract

Deep generative models often perform poorly in real-world applications due to the heterogeneity of natural data sets. Heterogeneity arises from data containing different types of features (categorical, ordinal, continuous, etc.) and features of the same type having different marginal distributions. We propose an extension of variational autoencoders (VAEs) called VAEM to handle such heterogeneous data. VAEM is a deep generative model that is trained in a two stage manner such that the first stage provides a more uniform representation of the data to the second stage, thereby sidestepping the problems caused by heterogeneous data. We provide extensions of VAEM to handle partially observed data, and demonstrate its performance in data generation, missing data prediction and sequential feature selection tasks. Our results show that VAEM broadens the range of real-world applications where deep generative models can be successfully deployed.

## 1 Introduction

Variational Autoencoders (VAEs) [15] are highly flexible probabilistic models, making them promising tools for enabling automated decision making under uncertainty in real-life scenarios. They are typically applied in standard settings in which each data dimension has a similar type and similar statistical properties (e.g. consider the pixels of an image). However, many real-world datasets contain variables with different types. For instance, in healthcare applications, a patient record may contain demographic information such as nationality which is of categorical type, the age which is ordinal, the height which is continuous.

Naively applying vanilla VAEs to such mixed type heterogeneous data can lead to unsatisfying results. The reason for this is that it requires the use of different likelihood functions (e.g. Gaussian likelihoods for real-valued variables and Bernoulli likelihoods for binary variables). In this case, the contribution that each likelihood makes to the training objective can be very different, leading to challenging optimization problems [13] in which some data dimensions may be poorly-modeled in favor of others. Figure 1 (c) shows an example in which a vanilla VAE fits some of the categorical variables, but performs poorly on the continuous ones.

To overcome the limitations of VAEs in this setting, we propose Variational Auto-encoder for heterogeneous mixed type data (VAEM) and study its performance for decision making in real-world

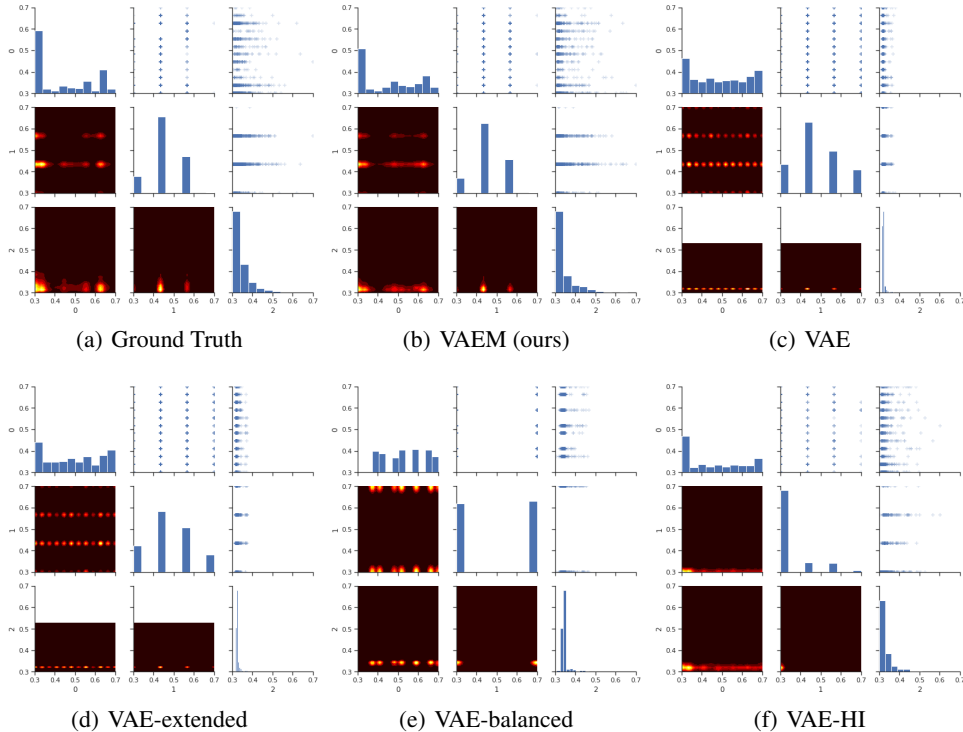

Figure 1: Pair plots of 3-dimensional data generated from 5 different models (defined in Section 5.1) and actual Bank data used to train them. In each subfigure, diagonal plots show marginal histograms for each variable. VAEM can correctly capture both continuous and discrete variables correctly both in terms of marginal distribution (diagonal plots) and pair-wise dependces (off-diagnal plots).

applications. VAEM uses a hierarchy of latent variables which is fit in two stages. In the first stage, we learn one type-specific VAE for each dimension. These initial one-dimensional VAEs capture marginal distribution properties and provide a latent representation that is more homogeneous across dimensions. In the second stage, another VAE is used to capture dependencies among the one-dimensional latent representations from the first stage.

Our contributions are the following:

- We present VAEM, a novel deep generative model for heterogeneous mixed-type data which alleviates the limitations of VAEs discussed above (See Section 2). We study the data generation quality of VAEM comparing with several existing VAE baselines on 5 different datasets. Our results show that VAEM can model mixed-type data more successfully than other baselines.

- We extend VAEM to handle missing data and perform conditional data generation, and derive algorithms that enable it to be used for sequential active information acquisition (Section 2.3). We show that VAEM obtains strong performance for conditional data generation as well as sequential active information acquisition in cases where VAEs perform poorly.

## 2  VAE for heterogeneous mixed type data

In this section, we first review VAEs and their naive application to heterogeneous mixed-typed data. Then, we describe our proposed VAEM, a two stage model developed for such heterogeneous mixed type data, and the corresponding amortized inference method. Finally, we briefly discuss connecions of VAEM with variational lower bounds and with data standardization methods.

## 2.1 Background: variational auto-encoders

Variational autoencoders (VAEs) [15, 25, 32] employ deep generative latent variable models that are trained using amortized variational inference. As shown in Figure 2(a), the VAE model assumes that the observed data $\mathbf{x}$ are generated from latent variable $\mathbf{z}$. The model is defined as $p_\theta(\mathbf{x}_n, \mathbf{z}_n) = p_\theta(\mathbf{x}_n|\mathbf{z}_n)p(\mathbf{z}_n)$. Here, $p_\theta(\mathbf{x}_n|\mathbf{z}_n)$ is often realized by a neural network known as the *decoder*. To approximate the posterior $p_\theta(\mathbf{z}_n|\mathbf{x}_n)$, VAEs use an encoder for *amortized inference*, which takes the data $\mathbf{x}_n$ as input to produce the variational parameters of the approximate posterior $q_\phi(\mathbf{z}_n|\mathbf{x}_n)$. Finally, VAEs can be trained by optimizing the variational lower bound (ELBO).

With VAEs, the likelihood is typically fully factorized, thus $p(\mathbf{x}_n|\mathbf{z}_n) = \prod_d p_d(x_{nd}|\mathbf{z}_n)$. In most machine learning applications, such as modeling images, each dimension of $\mathbf{x}_n$ has the same type, hence, each of these likelihood terms will take the same form, e.g. Gaussian.

**VAE for mixed type data**  A naive approach to handling heterogeneous mixed-typed data is to take the VAE model and use an appropriate likelihood function for each variable type. As discussed in Section 1, mixed likelihoods can cause problems in vanilla VAEs, causing them to perform poorly. A common choice in this case is to introduce a reweighting parameter for each dimension, which reweighs features from different type (Appendix D.1.2). However, it is not intuitive on how to set these temperature to obtain good performance.

## 2.2 VAE for heterogeneous mixed type data

In order to properly handle mixed type data with hetero-geneous marginals, our proposed method fits the data in a two-stage process. As shown in Figure 2(b), in the first stage we fit a different VAE independently to each data dimension $x_{nd}$. We call the resulting $D$ models *marginal VAEs*. Then, in the second stage, in order to capture the inter-variable dependencies, a new multi-dimensional VAE, called the *dependency network*, is build on top of the latent representations provided by the first-stage encoders. $D$ denotes the dimension of the observations and $N$ the number of data points with $x_{nd}$ being the $d$th dimension of the $n$th point. We present the details below.

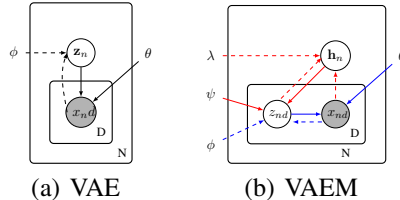

(a) VAE  (b) VAEM

Figure 2: Graphical representations of the vanilla VAE and our VAEM where solid arrows denote decoders, and dashed arrows are encoders.

***Stage one: training individual marginal VAEs to each single variable.*** In the first stage, we focus on modeling the marginal distributions of each variable by training $D$ individual VAEs $p_{\theta_d}(x_{nd}) = \mathbb{E}_{p(z_{nd})}p_{\theta_d}(x_{nd}|z_{nd})$, $\forall d \in \{1, 2, ..., D\}$ independently, i.e. each one is trained to fit a single dimension $x_{nd}$ from the dataset:

$$(\theta_d^\star, \phi_d^\star) = \arg\max_{\theta_d, \phi_d} \sum_n \mathbb{E}_{q_{\phi_d}(z_{nd}|x_{nd})} \log \frac{p_{\theta_d}(x_{nd}, z_{nd})}{q_{\phi_d}(z_{nd}|x_{nd})} \; \forall d \in \{1, 2, ..., D\}, \qquad (1)$$

where $p(z_{nd})$ is the standard Gaussian prior and $q_{\phi_d}(z_{nd}|x_{nd})$ is the Gaussian encoder of the $d$-th marginal VAE. To specify the likelihood terms $p_{\theta_d}(x_{nd}|z_{nd})$, we use Gaussian likelihoods for continuous data and categorical likelihoods with one-hot representation for categorical data. The case of other variable types is discussed in Appendix D.1.

Note that Equation 1 contains $D$ independent objectives. Each VAE $p_d(x_{nd}; \theta_d)$ is trained independently and is only responsible for modeling the individual statistical properties of a single dimension $x_{nd}$ from the dataset. Thus, we assume that $z_{nd}$ is a scalar without loss of generality, although it would be trivial to use a multi-dimensional $\mathbf{z}_{nd}$ instead. Each marginal VAE can be trained independently until convergence [1], hence avoiding the optimization issues of vanilla VAEs. We then fix the parameters of each marginal VAEs to be $\theta_d^\star$. These marginal VAEs fit the data well in practice, as shown in Figure 1, and have tight ELBOs, as discussed in Appendix E.2. Finally, note that the marginal VAE training is scalable to high dimensional data since the marginal VAEs can be trained in parallel. Thus, the computational complexity can be in the same order as VAE when sufficient compute resources are available. Moreover, the network size in each marginal VAE is very small as it only need to learn one dimensional marginal distribution for each feature. Our implementation is available at `https://github.com/microsoft/VAEM`.

***Stage two: training a dependency network to connect the marginal VAEs.*** In the second stage, we train a new multi-dimensional VAE on top of the latent representations $\mathbf{z}$ provided by the encoders of the first-stage marginal VAEs. This additional VAE, $p_\psi(\mathbf{z}) = \mathbb{E}_{p(\mathbf{h})} p_\psi(\mathbf{z}|\mathbf{h})$, models the inter-variable statistical dependencies and is called the *dependency network*. Here, $\mathbf{h}$ are the latent variables for the dependency network. Specifically, we train $p_\psi(\mathbf{z})$ as follows:

$$\mathbf{x}_n \sim p_{\text{data}}(\mathbf{x}), z_{nd} \sim q_{\phi_d}(z_d|x_{nd}), \quad \forall d \in \{1, ..., D\}, \tag{2}$$

$$(\psi^\star, \lambda^\star) \propto \arg\max_{(\psi, \lambda)} \sum_n \mathbb{E}_{q_\lambda(\mathbf{h}_n|\mathbf{z}_n, \mathbf{x}_n)} \log \frac{p_\psi(\mathbf{z}_n, \mathbf{h}_n)}{q_\lambda(\mathbf{h}_n|\mathbf{z}_n, \mathbf{x}_n)}. \tag{3}$$

The above procedure effectively disentangles the heterogeneous marginal properties of mixed type data (modelled by the marginal VAEs), from the inter-variable dependencies (modelled by the dependency network). We call our model *VAE for heterogeneous mixed type data (VAEM)*.

After training the marginal VAEs and dependency network, our final generative model is given by

$$p_\theta(\mathbf{x}) = \mathbb{E}_{(\mathbf{z}, \mathbf{h}) \sim p(\mathbf{h}) \prod_d p_\psi(z_d|\mathbf{h})} \left[ \prod_d p_{\theta_d}(x_d|z_d) \right]. \tag{4}$$

To handle complicated statistical dependencies, we use the VampPrior [30], which specifies a mixture of Gaussians (MoGs) as the prior distribution for the high-level latent variables, i.e., $p(\mathbf{h}) = \frac{1}{K} \sum_k q_\lambda(\mathbf{h}|\mathbf{u_k})$, where $K \ll N$ and the $\{\mathbf{u_k}\}$ are a subset of data points.

## 2.3 Discussions

**Optimization objective and relation to VAEs**  In Appendix A.2, we prove that the two stages of VAEM optimize a variational lower bound on the model likelihood $\sum_n \log \mathbb{E}_{p_\psi(\mathbf{z}_n)} \prod_d p_{\theta_d}(x_{nd}|z_{nd})$. In the first stage, we initialize $p_\psi(\mathbf{z}_n)$ to be a fully factorized standard Gaussian $p(\mathbf{z}_n)$ and keep it fixed, and optimize the rest of the parameters. This is obviously not an accurate prior since it does not consider dependencies among features. Thus, in the second stage, VAEM captures these dependencies by optimizing $p_\psi(\mathbf{z}_n)$ using the dependence VAE with variational distribution $q_\lambda(\mathbf{h}_n|\mathbf{z}_n, \mathbf{x}_n)$.

**VAEM as advanced data standardization**  In VAEM, the latent representations $z_d$ provided by the marginal VAEs are "Gaussianized" in the sense that they have homogeneous properties across dimensions. Each of these latent representations $z_d$ is encouraged to be homogeneously distributed as the prior distribution $p(z_{nd})$, that is, as the standard Gaussian distribution. In this way, we sidestep the heterogeneous mixed-type problem and the dependency VAE can focus on dependencies among homogeneous representations.

# 3 VAEM for sequential active information acquisition

The sequential active information acquisition task (SAIA), described by [18], can be used to evaluate the capability of generative models for decision making under uncertainty. In SAIA, a generative model is used to predict the missing data and actively decide which are the most informative missing features to collect next. Besides its high practical relevance, SAIA is useful for evaluating our methods because it quantifies how well the model fits the data and how good the inference process is. *In a heterogeneous data setting*, if a model cannot handle heterogeneity well, the performance in the SAIA task could be significantly deteriorated. To apply VAEM to this task, we extend VAEM to handle missing data and estimate the Lindley information [16].

## 3.1 Problem formulation

Suppose that, for a data instance $\mathbf{x}$, $\mathbf{x}_O$ denotes the set of currently observed variables, and $\mathbf{x}_U$ the unobserved ones. We are interested in predicting a target variable $\mathbf{x}_\Phi \in \mathbf{x}_U$ of interest given corresponding observed features $\mathbf{x}_O$ ($\mathbf{x}_\Phi \cap \mathbf{x}_O = \varnothing$). In this setting, a key problem is *sequential active information acquisition* (SAIA): given a generative model $p(\mathbf{x}_O, \mathbf{x}_U)$, how to decide which variable $\mathbf{x}_i \subset \mathbf{x}_{U \setminus \Phi}$ to observe next, so that we optimally increase our knowledge (e.g., predictive ability) regarding $\mathbf{x}_\Phi$?

As discussed in [18], to solve this problem we must have: 1) a good generative model that can handle missing data, and can effectively generate samples from the conditional distribution $p(\mathbf{x}_U|\mathbf{x}_O)$; and 2)

the ability to estimate a reward function given by the Lindley information. We now present extensions of VAEM to fulfill these two requirements.

## 3.2 Partial dependency network for handling missing data

The amortized inference network of VAEM (Section 2.2) cannot handle partially observed data, since the number of observed variables $\mathbf{x}_O$ might vary across different data instances. Inspired by the Partial VAE [18], we apply a PointNet to build a partial inference network in the dependency VAE that infers $\mathbf{h}$ from partial observations.

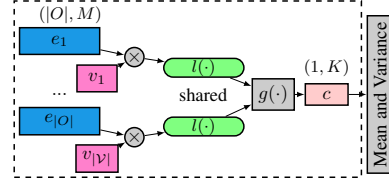

Figure 3: Illustration of the partial inference net for our dependency network.

During the first stage, we estimate each marginal VAE with only the observed samples for that dimension:

$$(\theta_d^\star, \phi_d^\star) = \arg\max_{\theta_d, \phi_d} \sum_n \mathbb{1}_{\{\mathbf{x}_{n,O}\}}(x_{nd}) \mathbb{E}_{q_{\phi_d}(z_{nd}|x_{nd})} \log \frac{p_{\theta_d}(x_{nd}, z_{nd})}{q_{\phi_d}(z_{nd}|x_{nd})}, \quad \forall d \in \{1, 2, ..., D\},$$

where $\mathbb{1}_{\{\mathbf{x}_{n,O}\}}(x_{nd})$ takes value one iff $x_{nd} \in \mathbf{x}_{n,O}$ and zero otherwise, with $\mathbf{x}_{n,O}$ being the set of observed variables for the $n$-th data instance.

For the second stage, we need a dependency VAE that can handle partial observations. Similarly as in the partial-VAE [18], in the presence of missing data, the dependency VAE specifies $p_{\mathbf{z}_O}(\mathbf{z}_O; \psi) = \mathbb{E}_{p(\mathbf{h})} \prod_{d \in O} p_\psi(z_d|\mathbf{h})$. This dependency VAE is trained by maximizing the partial ELBO:

$$\mathbb{E}_{q_\lambda(\mathbf{h}|\mathbf{z}_O, \mathbf{x}_O)} \log \frac{\prod_{d \in O} p_\psi(z_d, \mathbf{h}) p(\mathbf{h})}{q_\lambda(\mathbf{h}|\mathbf{z}_O, \mathbf{x}_O)}, z_d \sim q_d(z_d|x_{\text{data},d}, \phi_d) \; \forall d \in O, \;\; \mathbf{z}_O = \{z_d | d \in O\} \quad (5)$$

where $\mathbf{h}$ is the latent variable of the dependency network, $q_\lambda(\mathbf{h}|\mathbf{z}_O, \mathbf{x}_O)$ is a set-function, the so-called *partial inference net*, the structure of which is shown in Figure 3. Essentially, for each feature in $\mathbf{x}_O$, the input to the partial inference net is first modified as $\mathbf{s}_O := \{v \times \mathbf{e}_v | v \in \mathbf{z}_O \cup \mathbf{x}_O\}$ using element-wise multiplication, and $\mathbf{e}_v$ is a *feature embedding*[2]. $\mathbf{s}_O$ is then fed into a *feature map* (a neural network) $l(\cdot) : \mathbb{R}^M \to \mathbb{R}^K$, where $M$ and $K$ is the dimension of the feature embedding and the feature map, respectively. Finally, we apply a permutation invariant aggregation operation $g(\cdot)$, such as summation. In this way, $q_\lambda(\mathbf{h}|\mathbf{z}_O, \mathbf{x}_O)$ is invariant to the permutations of the elements of $\mathbf{x}_O$, and $\mathbf{x}_O$ can have arbitrary length.

**Approximate conditional data generation** Once the marginal VAEs and the partial dependency network are trained, we can generate conditional samples that approximate $p_\theta(\mathbf{x}_U|\mathbf{x}_O)$ by the following inference procedure: first, the latent representations $z_d$ for the observed variables are inferred. With this representation, we use the partial inference network to infer $\mathbf{h}$, which is the latent code for the second stage VAE. Given $\mathbf{h}$, we can generate the $z_s$ which are the latent code for the unobserved dimensions and then generate the $x_s$:

$$z_d \sim q_d(z_d|x_d, \phi_d) \; \forall d \in O, \;\; \mathbf{z}_O = \{z_d | d \in O\}, \mathbf{h} \sim q_\lambda(\mathbf{h}|\mathbf{z}_O, \mathbf{x}_O), z_s \sim p_\psi(z_s|\mathbf{h}) \; \forall s \in U,$$
$$\mathbf{z}_U = \{z_s | s \in U\}, x_s \sim p_\theta(x_s|z_s) \; \forall s \in U, \;\; \mathbf{x}_U = \{x_s | s \in U\}. \quad (6)$$

## 3.3 Reward estimation with VAEM

Following [18], SAIA can be framed as a Bayesian experimental design problem. The next variable to observe, $\mathbf{x}_i \subset \mathbf{x}_{U \setminus \Phi}$, is selected by the following *information reward* function:

$$R_I(\mathbf{x}_i, \mathbf{x}_O) = \mathbb{E}_{\mathbf{x}_i \sim p(\mathbf{x}_i|\mathbf{x}_O)} \mathbb{KL}\left[p(\mathbf{x}_\Phi|\mathbf{x}_i, \mathbf{x}_O) \,\|\, p(\mathbf{x}_\Phi|\mathbf{x}_O)\right].$$

Where $\mathbb{KL}$ is the Kullback-Leibler divergence. Intuitively this reward function selects a variable if it can result in the most drastic change in our current belief on $\mathbf{x}_\Phi$. Such change is captured by $\mathbb{KL}\left[p(\mathbf{x}_\Phi|\mathbf{x}_i, \mathbf{x}_O) \,\|\, p(\mathbf{x}_\Phi|\mathbf{x}_O)\right]$.

We use a trained partial VAEM model (6) to estimate the required distributions $p(\mathbf{x}_i|\mathbf{x}_O)$, $p(\mathbf{x}_\Phi|\mathbf{x}_i, \mathbf{x}_O)$, and $p(\mathbf{x}_\Phi|\mathbf{x}_O)$. Due to the intractability of $\mathbb{KL}\left[p(\mathbf{x}_\Phi|\mathbf{x}_i, \mathbf{x}_O) \| p(\mathbf{x}_\Phi|\mathbf{x}_O)\right]$, we must resort to approximations. An efficient estimation of $R_I(\mathbf{x}_i, \mathbf{x}_O)$ was proposed by [18], where the computations are reduced to a series of KL divergences in latent space. However, it cannot be applied in our case since our model is hierarchical and contains latent variables $\{\mathbf{z}_U\}$ with variable size due to missingness from $\mathbf{x}_U$. We hereby extend their method and show that $R_I(\mathbf{x}_i, \mathbf{x}_O)$ can be approximated as follows (Appendix A.1):

$$\hat{R}_I(\mathbf{x}_i, \mathbf{x}_O) = \mathbb{E}_{p_\theta(\mathbf{x}_i, \mathbf{z}_i, \mathbf{z}_O|\mathbf{x}_O)} \left\{ \mathbb{KL}\left[q_\lambda(\mathbf{h}|\mathbf{z}_i, \mathbf{z}_O)||q_\lambda(\mathbf{h}|\mathbf{z}_O)\right] - \right. \tag{7}$$
$$\left. \mathbb{E}_{p_\theta(\mathbf{x}_\phi, \mathbf{z}_\Phi,|\mathbf{x}_O)} \mathbb{KL}\left[q_\lambda(\mathbf{h}|\mathbf{z}_\Phi, \mathbf{z}_i, \mathbf{z}_O)||q_\lambda(\mathbf{h}|\mathbf{z}_\Phi, \mathbf{z}_O)\right]\right\}.$$

Note that, for compactness, we omitted the inputs $\mathbf{x}_O$ and $\mathbf{x}_i$ to the partial inference networks. The approximation (7) is very efficient to compute, since all KL terms can be calculated analytically, assuming that the partial inference net $q_\lambda(\mathbf{h}|\mathbf{z}_O)$ is Gaussian (or other common distributions where KL divergences can be estimated deterministically or via Monte Carlo).

### 3.4 Enhancing the predictive performance of VAEM

To predict the variable of interest is desirable to use a supervised learning method instead of just an unsupervised method such as the VAE. In active information acquisition, the target of interest $\mathbf{x}_\Phi$ ($\mathbf{x}_\Phi \cap \mathbf{x}_O = \varnothing$) is often the variable that we try to predict in a regression/classification task. In order to enhance the predictive performance of VAEM, we propose to use the following factorization:

$$p_\theta(\mathbf{x}_O, \mathbf{x}_\Phi) = \mathbb{E}_{p_\theta(\mathbf{x}_{U\setminus\Phi}, \mathbf{h}|\mathbf{x}_O)} p_\gamma(\mathbf{x}_\Phi|\mathbf{x}_O, \mathbf{x}_{U\setminus\Phi}, \mathbf{h}) p_\theta(\mathbf{x}_O), \tag{8}$$

where $p_\gamma(\mathbf{x}_\Phi|\mathbf{x}_O, \mathbf{x}_{U\setminus\Phi}, \mathbf{h})$ is the discriminator (prediction model) that takes the observed variables $\mathbf{x}_O$, the imputed variables $\mathbf{x}_{U\setminus\Phi}$ and the global latent representation $\mathbf{h}$ as input, and predicts the distribution of $\mathbf{x}_\Phi$. Training the joint model in Equation 8 is similar to our previous two-stage procedure, which is detailed in Appendix C.

## 4 Related works

VAEM is a two stage method that extends generative models to handle mixed type heterogeneous data, with applications in SAIA. Therefore, we review here the literature in the following aspects.

**Generative models for mixed-type heterogeneous data** This type of models are under-explored in the literature. [22] proposed Heterogeneous-Incomplete VAE (HI-VAE), a deep generative model with heterogeneous variables. It uses a multi-head decoder architecture [23, 6]. However, this does not help balance the learning of different marginal distributions. A recent empirical study [17] shows that HI-VAE fails to recover the marginal distributions correctly. Finally, another orthogonal line of work focuses on using traditional latent variable models to infer the variable types automatically [4, 3, 31, 8]. However, they have been shown to be surpassed by VAE-based models empirically [22].

**VAEs for Sequential Active Information Acquisition** There has been a recent drive to probabilistic methods for sequential active information acquisition (SAIA) [18, 5]. SAIA differs from traditional active learning (AL) [16, 19, 27] settings in that AL assumes access to fully observed unlabelled data, and performs *instance-wise* selection for best predictive performance during training, whilst SAIA performs *variable-wise acquisition* for each data instance at test-time. Thus, SAIA are of great importance in many real-world applications where every feature from every data point is associated with cost. SAIA is often performed with the help of generative models to describe the underlying data generation process. For example, in the recently proposed EDDI solution for SAIA tasks [5, 18], the partial VAE is proposed to estimate all conditional distributions required by SAIA. Both [18, 5] are designed for data with only continuous variables. Our work extends their work and demonstrates how to use VAEM in the same setting as EDDI and the same way can be applied to Icebreaker [5]. Apart from SAIA, there also exist many closely related methods in the field of active feature acquisition (AFA) for classification [20, 26, 29, 9]. In particular, SAIA can be viewed as a specific case of AFA applied to prediction time[12, 11, 28]. From this perspective, AFA methods can also potentially benefit from our work in scenarios where heterogeneous data is present, by using VAEM as part of the data imputation module for classifiers.

**Two-stage methods for VAEs** An orthogonal line of work (TS-VAE) [1], uses a two-stage training paradigm to improve VAEs when applied to homogeneous continuous data. First, a VAE is trained on

Table 1: Data generation quality in terms of average test NLL per variable and corresponding standard errors across different datasets.

| Method | Ours | VAE | VAE-balanced | VAE-extended | VAE-HI |
|---|---|---|---|---|---|
| Bank | **-1.15±.09** | 2.09±.04 | 0.72±.01 | 2.06±.00 | -0.72±.00 |
| Boston | **-2.16±.01** | -1.69±.01 | 0.38±.01 | -1.61±.02 | 2.11±.01 |
| Avocado | **-0.16±.00** | 0.04±.00 | 1.32±.01 | 0.04±.00 | 0.04±.00 |
| Energy | -1.28±.09 | **-1.47±.07** | 0.69±.02 | -1.46±.08 | 0.16±.00 |
| MIMIC | **-1.01±.00** | 0.08±.00 | 0.69±.00 | 0.08±.00 | 0.08±.00 |
| Avg. Rank | **1.40±.36** | 2.60±.61 | 4.40±.36 | 3.00±.40 | 3.00±.57 |

Table 2: Conditional data generation quality under random missing entries. Average test NLL per variable and corresponding standard errors across different datasets.

| Method | Ours | VAE | VAE-balanced | VAE-extended | VAE-HI |
|---|---|---|---|---|---|
| Bank | **-1.21±.12** | 2.09±.00 | 0.68±.00 | 2.09±.00 | -0.83±.01 |
| Boston | **-2.18±.03** | -1.66±.02 | 0.37±.00 | -1.67±.01 | 1.58±.01 |
| Avocado | **-0.15±.00** | 0.04±.00 | 1.33±.00 | 0.04±.00 | 0.04±.00 |
| Energy | -1.30±.05 | **-1.50±.06** | 0.67±.01 | -1.50±.06 | 0.13±.00 |
| MIMIC | **-1.10±.00** | 0.08±.00 | 0.57±.00 | 0.08±.00 | 0.08±.00 |
| Avg. Rank | **1.40±.36** | 2.60±.61 | 4.40±.38 | 2.30±.44 | 3.00±.57 |

the data to discover low-dimensional latent representations; Then, another VAE is trained on these representations to capture any deviations from the prior in the latent space of the first VAE. In this regard, this method is similar to the VampPrior VAE [30], where a mixture of Gaussians is optimized to provide a flexible prior on the VAE latent variables. However, unlike our approach, both TS-VAE and VampPrior do not naturally handle mixed type data and missing data.

## 5 Experiments

In this section, we evaluate the performance and validity of VAEM. We focus on three different tasks with mixed type heterogeneous data: 1) data modeling and generation, 2) conditional data generation (imputation) and 3) sequential active information acquisition. For consistency, we use the same set of baselines and datasets across different tasks whenever possible.

### 5.1 Baselines and datasets

In the experiments, we consider a number of baselines. Unless otherwise specified, all VAE baselines use the partial inference network and the discriminator specified in Section 3.2 and 3.4, respectively. Moreover, all baselines are equipped with a MoG priors (Section 2.2). Our main baselines include:

- Heterogeneous-Incomplete VAE [22]. We adopt the multi-head structure of HI-VAE and match the dimensionality of latent variables to be the same as in VAEM. HI-VAE is an important baseline, since it is motivated in a similar way as our VAEM, but all marginal VAEs are trained jointly rather as opposed to our two-stage method. In addition, HI-VAE uses deterministic units for $\mathbf{z}$. We denote this baseline by `VAE-HI`.
- VAE: A vanilla VAE equipped with a VampPrior [30]. The number of latent dimensions is the same as in the second stage ($\mathbf{h}$) of VAEM. We denote this by `VAE`.
- VAE with extended latent dimension: same as the `VAE`, but with the latent dimension increased to be the same as VAEM (sum of the dimensions of $\mathbf{h}$ and $\mathbf{z}$). We denote this by `VAE-extended`.
- VAE with balanced likelihoods. This baseline automatically equal the scale of each likelihood term of the different variable types, by multiplying each likelihood term with an adaptive constant (Appendix D.1). We denote this baseline by `VAE-balanced`.

We use the same collection of mixed type datasets in all tasks:

- Two standard UCI benchmark datasets: Boston housing and energy efficiency [2];

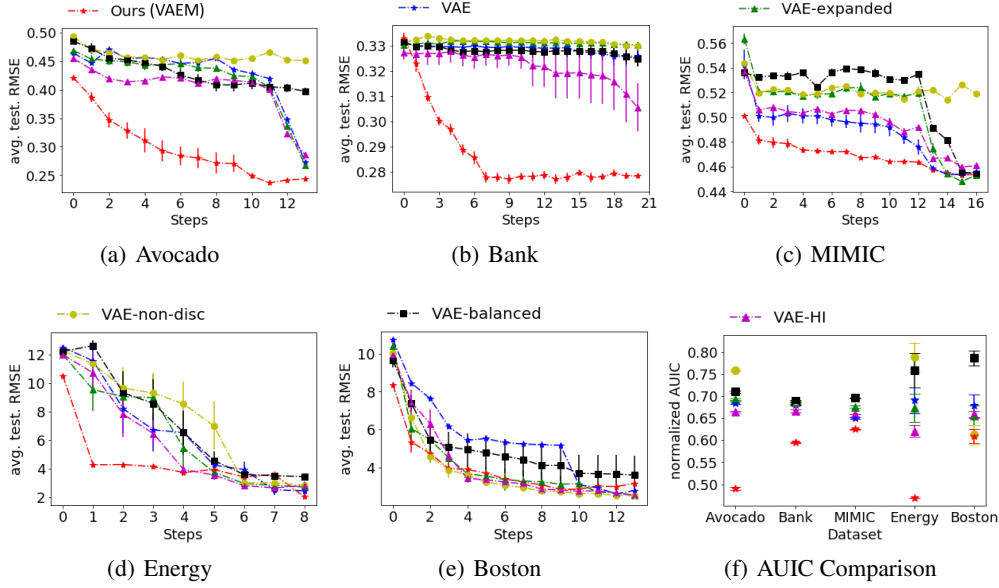

(a) Avocado     (b) Bank     (c) MIMIC

(d) Energy     (e) Boston     (f) AUIC Comparison

Figure 4: Information curves of sequential active information acquisition, with standard error as error bars. **((a)-(e))**: Information curves of Avocado sales, Bank marketing, MIMIC-III, Energy, Boston Housing, respectively. **(f)**: Comparison of AUIC on each dataset. All AUIC values are normalized by dividing by the average AUIC value within the corresponding dataset.

- Two relatively large real-world datasets: Bank marketing;[21] and Avocado sales prediction.
- A real-world medical dataset: MIMIC III [10], the largest public medical dataset for intensive care. We focus on the mortality prediction task.

Details including model details, hyperparameters and data processing can be found in Appendix D.

## 5.2 Mixed type data generation

In this task, we evaluate the quality of our generative model in terms of mixed type data generation. For all datasets, we first train the models and then quantitatively compare their performance using a 90%-10% train-test split. All experiments are repeated 5 times with different random seeds.

**Visualization by pair plots** In deep generative models, the data generation quality is indicative of how well the model describes the data. Thus, we first visualize the data generated by each model on a representative dataset: Bank marketing. This dataset contains three different data types with drastically different marginals, which present a challenge for learning. We fit our models to the Bank data and then generate pair plots for three variables, $x_0$, $x_1$ and $x_2$ (the first two are categorical and the third one is continuous), selected from the data (see Figure 1). Full plots with all the other variables can be found in Appendix F. In each subfigure of Figure 1, diagonal plots show the marginal histograms of each variable. The upper-triangular part shows sample scatter plots for each variable pair. The lower-triangular part shows heat maps identifying regions of high-probability density for each variable pair, as given by kernel density estimates.

By comparing the plots in the diagonals of Figure 1 (a) and Figure 1 (c), we notice that vanilla VAE is able to describe the marginal distribution of the second categorical variable. However, it fails to mimic the behaviour of the third variable. Note that this variable (Figure 1 (a)), which corresponds to the "duration" feature of the dataset, has a heavy tail behaviour, which is ignored by vanilla VAE. On the other hand, although the VAE-balanced model and VAE-HI (Figures 1 (e) and (f)) can partially describe this heavy-tail behaviour, they fail to model the marginal distribution of the second categorical variable well. Unlike the baselines, our VAEM model (Figure 1 (b)) is able to accurately describe the marginals and joint distributions for both categorical and heavy-tailed continuous distributions.

**Quantitative evaluation on all datasets**  To evaluate the data generation quality quantitatively, we compute the average negative log-likelihood (NLL) of the models on the test sets (detailed in Appendix D.2). Note that all NLL numbers are divided by the actual number of variables in each dataset. As shown in Table 1, VAEM can consistently provide a very good fit of the data, and on average significantly outperforms the other baselines.

## 5.3  Mixed type conditional data generation

An important aspect of generative models is their ability to perform conditional data generation. That is, given a data instance, to infer the posterior distribution of unobserved variables $\mathbf{x}_U$ given observed $\mathbf{x}_O$. For all baselines evaluated in this task, we train the partial version of them (i.e., generative model + partial inference net [18]). We manually drops 50% of the data entries from test set for imputation. Since all inference methods are probabilistic, we report the average test NLLs on unobserved data, as opposed to the imputation error, which is more typically used in the literature. In addition to NLLs, we also provide imputation error in term of RMSE in Appendix E.1. The results are consistent and our proposed VAEM consistently show improved performance over different baseline methods.

Results are summarized in Table 2, where all NLL values have been divided by the number of observed variables. We repeat our experiments for 5 runs. Note that the automatic balancing strategy **VAE-balanced** almost always deteriorates performance. By contrast, Table 2 shows that our proposed method is very robust, yielding significant improvements over all baselines on 4 out of 5 datasets.

## 5.4  Sequential active information acquisition (SAIA)

In our final experiments, we apply VAEM to the task of sequential active information acquisition (SAIA) based on the formulation described in Section 2.3. We use this task as an example to showcase how VAEM can be used in decision making under uncertainty.

We employ the same experiment pipeline as in [18]. The reward function of VAEM is estimated according to Section 2.3. We add an additional baseline, denoted by `VAE-no-disc`, where the prediction of the target is directly generated by the decoder without using a predictive model. By comparing to this baseline, we can show the importance of the discriminator described in Section 3.4. The other settings are the same as described in Section 5.1. All baseline methods use the information reward estimation method proposed in [18]. All experiments are repeated ten times.

Figure 4 shows the average test RMSEs on $\mathbf{x}_\Phi$ for each variable selection step on all five datasets, where $\mathbf{x}_\Phi$ is the target variable. The y-axis shows the error of the prediction and the x-axis shows the number of features acquired so far. The curves in Figure 4 are called information curves [18, 5]. The area under the information curve (AUIC) can be used to evaluate the performance of a method in SAIA. The smaller the area, the better the method. From Figure 4, we see that VAEM performs consistently better than the other baselines. Note that, on the Bank marketing and Avocado sales datasets, a lot of heterogeneous variables are involved and other baselines fail to reduce the test RMSE quickly and VAEM outperforms them by a large margin. These experiments show that VAEM is able to acquire information efficiently on mixed type datasets.

## 6  Conclusion

We proposed VAEM, a novel two stage deep generative model that can handle mixed type data with heterogeneous marginals and missing data. VAEM sidesteps the problems arising from fitting heterogeneous data directly. For this, VAEM uses a two-stage training procedure. Efficient amortized inference methods and extensions are proposed. Experiments yield promising results, indicating that VAEM is useful for real-world applications of deep generative models.

Many real-world applications are associated with large scale mixed-type tabular data. In this paper, we have mainly examined the performance on typical machine learning benchmarks. In future works, we will apply this method to other real-life settings and further extend the VAEM method. In addition, when using VAEM for missing value imputation, we currently assume that the data are missing completely at random (MCAR). However, in many scenarios, data are missing not at random (MNAR). We would thus like to continue our research on missing data imputation for heterogeneous mixed type data under all different missingness mechanisms.

## Broader Impact

Our paper provides a simple solution for handling heterogeneous mixed-type data with deep generative models. This is an under explored topic in the literature, which to a certain extend, limits the application of deep generative models to real-life tasks. Our research opens-up new possibilities to VAEs, and broadens the range of real-world applications where deep generative models can be successfully deployed.

As a model, we foresee that our research is a useful addition to the deep generative modeling toolbox. It is of particular interest to ML researchers and practitioners, whenever a good probabilistic model for mixed-type data is needed. Our SAIA algorithm and data imputation algorithm with VAEM is also valuable for ML practitioners who is looking for immediate solutions for their specific scenarios at hand.

## Acknowledgements and Disclosure of Funding

We thank Yingzhen Li for useful discussions. The authors thank Microsoft Research Cambridge for financial support of this work.

## Footnotes

*This work was performed when the authors were (part-time) associated with Microsoft Research, Cambridge

[2]If $v$ is a non-continuous variable such as categorical, the operation $v \times \mathbf{e}_v$ is performed on the one-hot representation of $v$, as detailed in Appendix D.1

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
