[Supplementary Material]

# A Additional Derivations

## A.1 Information reward approximation for hierarchical generative models in the present of missing latent variable

We consider the estimation of the following *information reward* function

$$R_I(\mathbf{x}_i, \mathbf{x}_O) = \mathbb{E}_{\mathbf{x}_i \sim p(\mathbf{x}_i|\mathbf{x}_O)} \mathbb{KL}\left[p(\mathbf{x}_\Phi|\mathbf{x}_i, \mathbf{x}_O) \,\|\, p(\mathbf{x}_\Phi|\mathbf{x}_O)\right]$$

Using our proposed VAEM method (the partial VAEM version in 3.2). The VAEM is a hierarchical generative model trained by the two-stage procedure described in the paper. Conditional inference of VAEM of missing data follows the following sampling process:

$$
\begin{aligned}
z_d &\sim q_d(z_d|x_{,d}, \boldsymbol{\Phi}_d) \ \ \forall d \in O, \ \ \mathbf{z}_O = \{z_d | d \in O\} \\
\mathbf{h} &\sim q_\lambda(\mathbf{h}|\mathbf{z}_O) \\
z_s &\sim p_\psi(z_s|\mathbf{h}) \ \ \forall s \in U, \ \ \mathbf{z}_U = \{z_s | s \in U\} \\
x_s &\sim p_\theta(x_s|\mathbf{z}_U, \mathbf{z}_O) \ \ \forall s \in U, \ \ \mathbf{x}_U = \{x_s | s \in U\}
\end{aligned}
$$

Note that for compactness, we omitted the notation for input $\mathbf{x}_O$ and $\mathbf{x}_i$ to the all partial inference nets $q_\lambda$. Where $\mathbf{z}_O$ is the observed latent variables of marginal VAEs, and $\mathbf{z}_U$ are unobserved. We will use this VAEM to estimate any probabilistic quantities in information reward A.1.

Applying the chain rule of KL-divergence on the term $\mathbb{KL}\left[p(\mathbf{x}_\Phi|\mathbf{x}_i, \mathbf{x}_O) \,\|\, p(\mathbf{x}_\Phi|\mathbf{x}_O)\right]$, we have:

$$
\begin{aligned}
&\mathbb{KL}(p(\mathbf{x}_\Phi|\mathbf{x}_i, \mathbf{x}_O)||p(\mathbf{x}_\Phi|\mathbf{x}_O)) \\
&= \mathbb{KL}(p(\mathbf{x}_\Phi, \mathbf{z}_i, \mathbf{z}_O, \mathbf{h}|\mathbf{x}_i, \mathbf{x}_O)||p(\mathbf{x}_\Phi, \mathbf{z}_i, \mathbf{z}_O, \mathbf{h}|\mathbf{x}_O)) \\
&\quad - \mathbb{E}_{\mathbf{x}_\Phi \sim p(\mathbf{x}_\Phi|\mathbf{x}_i, \mathbf{x}_O)}\left[\mathbb{KL}(p(\mathbf{z}_\Phi, \mathbf{z}_i, \mathbf{z}_O, \mathbf{h}|\mathbf{x}_\Phi, \mathbf{x}_i, \mathbf{x}_O)||p(\mathbf{z}_\Phi, \mathbf{z}_i, \mathbf{z}_O, \mathbf{h}|\mathbf{x}_\Phi, \mathbf{x}_O))\right],
\end{aligned}
$$

Based on the independencies of marginal VAEs, we have $p(\mathbf{x}_\Phi, \mathbf{z}_i, \mathbf{z}_O, \mathbf{h}|\mathbf{x}_O)) = p(\mathbf{x}_\Phi, \mathbf{z}_O, \mathbf{h}|\mathbf{x}_O))p(\mathbf{z}_i), p(\mathbf{z}_\Phi, \mathbf{z}_i, \mathbf{z}_O, \mathbf{h}|\mathbf{x}_\Phi, \mathbf{x}_O)) = p(\mathbf{z}_\Phi, \mathbf{z}_i, \mathbf{z}_O, \mathbf{h}|\mathbf{x}_\Phi, \mathbf{x}_O))p(\mathbf{z}_i)$.

Using again the KL-divergence chain rule on $\mathbb{KL}(p(\mathbf{x}_\Phi, \mathbf{z}_i, \mathbf{z}_O, \mathbf{h}|\mathbf{x}_i, \mathbf{x}_O)||p(\mathbf{x}_\Phi, \mathbf{z}_i, \mathbf{z}_O, \mathbf{h}|\mathbf{x}_O))$, we have:

$$
\begin{aligned}
&\mathbb{KL}(p(\mathbf{x}_\Phi, \mathbf{z}_i, \mathbf{z}_O, \mathbf{h}|\mathbf{x}_i, \mathbf{x}_O)||p(\mathbf{x}_\Phi, \mathbf{z}_i, \mathbf{z}_O, \mathbf{h}|\mathbf{x}_O)) \\
&= \mathbb{KL}(p(\mathbf{z}_i, \mathbf{z}_O, \mathbf{h}|\mathbf{x}_i, \mathbf{x}_O)||p(\mathbf{z}_i, \mathbf{z}_O, \mathbf{h}|\mathbf{x}_O)) + \mathbb{E}_{p(\mathbf{z}_\Phi, \mathbf{z}_i, \mathbf{z}_O, \mathbf{h}|\mathbf{x}_i, \mathbf{x}_O))}\mathbb{KL}(p(\mathbf{x}_\Phi|\mathbf{z}_i, \mathbf{z}_O, \mathbf{h}, \mathbf{x}_i, \mathbf{x}_O)||p(\mathbf{x}_\Phi|\mathbf{z}_i, \mathbf{z}_O, \mathbf{h}, \mathbf{x}_O)) \\
&= \mathbb{KL}(p(\mathbf{z}_i, \mathbf{z}_O, \mathbf{h}|\mathbf{x}_i, \mathbf{x}_O)||p(\mathbf{z}_i, \mathbf{z}_O, \mathbf{h}|\mathbf{x}_O)) + \mathbb{E}_{p(\mathbf{z}_\Phi, \mathbf{z}_i, \mathbf{z}_O, \mathbf{h}|\mathbf{x}_i, \mathbf{x}_O))}\mathbb{KL}(p(\mathbf{x}_\Phi|\mathbf{z}_i, \mathbf{z}_O, \mathbf{h})||p(\mathbf{x}_\Phi|\mathbf{z}_i, \mathbf{z}_O, \mathbf{h})) \\
&= \mathbb{KL}(p(\mathbf{z}_i, \mathbf{z}_O, \mathbf{h}|\mathbf{x}_i, \mathbf{x}_O)||p(\mathbf{z}_i, \mathbf{z}_O, \mathbf{h}|\mathbf{x}_O)).
\end{aligned}
$$

Note that the last two equalities does not hold for the discriminative version of VAEM described in Section 3.4. Fortunately, $\mathbb{E}_{\mathbf{x}_i \sim p(\mathbf{x}_i|\mathbf{x}_O)} \mathbb{KL}(p(\mathbf{x}_\Phi|\mathbf{z}_i, \mathbf{z}_O, \mathbf{h}, \mathbf{x}_i, \mathbf{x}_O)||p(\mathbf{x}_\Phi|\mathbf{z}_i, \mathbf{z}_O, \mathbf{h}, \mathbf{x}_O)) = 0$ still holds for the discriminative version, hence we will still arrive at the same result.

The KL-divergence term in the reward formula is now rewritten as follows,

$$
\begin{aligned}
&\mathbb{KL}(p(\mathbf{x}_\Phi|\mathbf{x}_i, \mathbf{x}_O)||p(\mathbf{x}_\Phi|\mathbf{x}_O)) \\
&= \textcolor{blue}{\mathbb{KL}(p(\mathbf{z}_i, \mathbf{z}_O, \mathbf{h}|\mathbf{x}_i, \mathbf{x}_O)||p(\mathbf{z}_i, \mathbf{z}_O, \mathbf{h}|\mathbf{x}_O))} \\
&\quad - \mathbb{E}_{\mathbf{x}_\Phi \sim p(\mathbf{x}_\Phi|\mathbf{x}_i, \mathbf{x}_O)}\left[\mathbb{KL}(p(\mathbf{z}_\Phi, \mathbf{z}_i, \mathbf{z}_O, \mathbf{h}|\mathbf{x}_\Phi, \mathbf{x}_i, \mathbf{x}_O)||p(\mathbf{z}_\Phi, \mathbf{z}_i, \mathbf{z}_O, \mathbf{h}|\mathbf{x}_\Phi, \mathbf{x}_O))\right].
\end{aligned}
$$

For the term in blue, we have:

$$
\begin{aligned}
&\textcolor{blue}{\mathbb{KL}(p(\mathbf{z}_i, \mathbf{z}_O, \mathbf{h}|\mathbf{x}_i, \mathbf{x}_O)||p(\mathbf{z}_i, \mathbf{z}_O, \mathbf{h}|\mathbf{x}_O))} \\
&= \mathbb{KL}(p(\mathbf{z}_i, \mathbf{z}_O|\mathbf{x}_i, \mathbf{x}_O)||p(\mathbf{z}_O|\mathbf{x}_O)p(\mathbf{z}_i)) \\
&\quad + \mathbb{E}_{\mathbf{z}_i, \mathbf{z}_O \sim p(\mathbf{z}_i, \mathbf{z}_O|\mathbf{x}_i, \mathbf{x}_O)}\left[\mathbb{KL}\left(p(\mathbf{h}|\mathbf{z}_i, \mathbf{z}_O)||p(\mathbf{h}|\mathbf{z}_O)\frac{p(\mathbf{z}_i)}{p(\mathbf{z}_i|\mathbf{x}_O)}\right)\right] \\
&= \mathbb{KL}(p(\mathbf{z}_i|\mathbf{x}_i)||p(\mathbf{z}_i)) + \mathbb{E}_{\mathbf{z}_i, \mathbf{z}_O \sim p(\mathbf{z}_i, \mathbf{z}_O|\mathbf{x}_i, \mathbf{x}_O)}\left[\mathbb{KL}\left(p(\mathbf{h}|\mathbf{z}_i, \mathbf{z}_O)||p(\mathbf{h}|\mathbf{z}_O)\right)\right]
\end{aligned}
$$

Similarly for the term in red, we have:

$$\mathbb{KL}(p(\mathbf{z}_\Phi, \mathbf{z}_i, \mathbf{z}_O, \mathbf{h}|\mathbf{x}_\Phi, \mathbf{x}_i, \mathbf{x}_O)||p(\mathbf{z}_\Phi, \mathbf{z}_i, \mathbf{z}_O, \mathbf{h}|\mathbf{x}_\Phi, \mathbf{x}_O))$$

$$= \mathbb{KL}(p(\mathbf{z}_\Phi, \mathbf{z}_i, \mathbf{z}_O|\mathbf{x}_\Phi, \mathbf{x}_i, \mathbf{x}_O)||p(\mathbf{z}_\Phi, \mathbf{z}_O|\mathbf{x}_\Phi, \mathbf{x}_O)p(\mathbf{z}_i))$$

$$+ \mathbb{E}_{\mathbf{z}_\Phi, \mathbf{z}_i, \mathbf{z}_O \sim p(\mathbf{z}_\Phi, \mathbf{z}_i, \mathbf{z}_O|\mathbf{x}_\Phi, \mathbf{x}_i, \mathbf{x}_O)} \left[ \mathbb{KL}\left( p(\mathbf{h}|\mathbf{z}_\Phi, \mathbf{z}_i, \mathbf{z}_O)||p(\mathbf{h}|\mathbf{z}_\Phi, \mathbf{z}_O)\frac{p(\mathbf{z}_i)}{p(\mathbf{z}_i|\mathbf{x}_\Phi, \mathbf{x}_O)} \right) \right]$$

$$= \mathbb{KL}(\mathbf{z}_i|\mathbf{x}_i)||p(\mathbf{z}_i)) + \mathbb{E}_{\mathbf{z}_\Phi, \mathbf{z}_i, \mathbf{z}_O \sim p(\mathbf{z}_\Phi, \mathbf{z}_i, \mathbf{z}_O|\mathbf{x}_\Phi, \mathbf{x}_i, \mathbf{x}_O)} [\mathbb{KL}(p(\mathbf{h}|\mathbf{z}_\Phi, \mathbf{z}_i, \mathbf{z}_O)||p(\mathbf{h}|\mathbf{z}_\Phi, \mathbf{z}_O))]$$

Finally, we have:

$$R_I(\mathbf{x}_i, \mathbf{x}_O)$$
$$= \mathbb{E}_{\mathbf{x}_i \sim p(\mathbf{x}_i|\mathbf{x}_O)} \mathbb{KL}\left[ p(\mathbf{x}_\Phi|\mathbf{x}_i, \mathbf{x}_O) \,\|\, p(\mathbf{x}_\Phi|\mathbf{x}_O) \right]$$
$$= \mathbb{E}_{\mathbf{x}_i \sim p(\mathbf{x}_i|\mathbf{x}_O)} \mathbb{KL}(p(\mathbf{z}_i, \mathbf{z}_O, \mathbf{h}|\mathbf{x}_i, \mathbf{x}_O)||p(\mathbf{z}_i, \mathbf{z}_O, \mathbf{h}|\mathbf{x}_O))$$
$$- \mathbb{E}_{\mathbf{x}_i \sim p(\mathbf{x}_i|\mathbf{x}_O)} \mathbb{E}_{\mathbf{x}_\Phi \sim p(\mathbf{x}_\Phi|\mathbf{x}_i, \mathbf{x}_O)} [\mathbb{KL}(p(\mathbf{z}_\Phi, \mathbf{z}_i, \mathbf{z}_O, \mathbf{h}|\mathbf{x}_\Phi, \mathbf{x}_i, \mathbf{x}_O)||p(\mathbf{z}_\Phi, \mathbf{z}_i, \mathbf{z}_O, \mathbf{h}|\mathbf{x}_\Phi, \mathbf{x}_O))]$$
$$= \mathbb{E}_{\mathbf{x}_i, \mathbf{z}_i, \mathbf{z}_O \sim p(\mathbf{x}_i, \mathbf{z}_i, \mathbf{z}_O|\mathbf{x}_O)} \{ \mathbb{KL}\left[ p(\mathbf{h}|\mathbf{z}_i, \mathbf{z}_O)||p(\mathbf{h}|\mathbf{z}_O) \right]$$
$$- \mathbb{E}_{\mathbf{x}_\Phi, \mathbf{z}_\Phi \sim p(\mathbf{x}_\Phi, \mathbf{z}_\Phi, |\mathbf{x}_O)} \mathbb{KL}\left[ p(\mathbf{h}|\mathbf{z}_\Phi, \mathbf{z}_i, \mathbf{z}_O)||p(\mathbf{h}|\mathbf{z}_\Phi, \mathbf{z}_O) \right] \} \,.$$

We can then plug in the VAEM model distirbutions:

$$p(\mathbf{x}_i, \mathbf{z}_i, \mathbf{z}_O|\mathbf{x}_O) = p_{\theta,\phi}(\mathbf{x}_i, \mathbf{z}_i, \mathbf{z}_O|\mathbf{x}_O)$$
$$p(\mathbf{x}_\Phi, \mathbf{z}_\Phi, |\mathbf{x}_O) = p_{\theta,\phi}(\mathbf{x}_\Phi, \mathbf{z}_\Phi, |\mathbf{x}_O)$$
$$p(\mathbf{h}|\mathbf{z}_i, \mathbf{z}_O) \approx q_\lambda(\mathbf{h}|\mathbf{z}_i, \mathbf{z}_O)$$
$$p(\mathbf{h}|\mathbf{z}_O) \approx q_\lambda(\mathbf{h}|\mathbf{z}_O)$$
$$p(\mathbf{h}|\mathbf{z}_\Phi, \mathbf{z}_i, \mathbf{z}_O) \approx q_\lambda(\mathbf{h}|\mathbf{z}_\Phi, \mathbf{z}_i, \mathbf{z}_O)$$
$$p(\mathbf{h}|\mathbf{z}_\Phi, \mathbf{z}_O) \approx q_\lambda(\mathbf{h}|\mathbf{z}_\Phi, \mathbf{z}_O)$$

Finally, the information reward is now approximated as:

$$R_I(\mathbf{x}_i, \mathbf{x}_O)$$
$$\approx \mathbb{E}_{\mathbf{x}_i, \mathbf{z}_i, \mathbf{z}_O \sim p_{\theta,\phi}(\mathbf{x}_i, \mathbf{z}_i, \mathbf{z}_O|\mathbf{x}_O)} \{ \mathbb{KL}\left[ q_\lambda(\mathbf{h}|\mathbf{z}_i, \mathbf{z}_O)||q_\lambda(\mathbf{h}|\mathbf{z}_O) \right]$$
$$- \mathbb{E}_{\mathbf{x}_\Phi, \mathbf{z}_\Phi \sim p_{\theta,\phi}(\mathbf{x}_\Phi, \mathbf{z}_\Phi, |\mathbf{x}_O)} \mathbb{KL}\left[ q_\lambda(\mathbf{h}|\mathbf{z}_\Phi, \mathbf{z}_i, \mathbf{z}_O)||q_\lambda(\mathbf{h}|\mathbf{z}_\Phi, \mathbf{z}_O) \right] \} \,.$$

## A.2 VAEM optimizes a lower bound of joint model log-likelihood

Next, we show that VAEM improves a valid lower bound of the true log-likelihood. Recall that in the first stage, $D$ individual VAEs are trained independently, i.e. each one is trained to fit a single dimension $x_{nd}$ from the dataset:

$$(\theta_d^\star, \phi_d^\star) = \arg\max_{\theta_d, \phi_d}$$
$$\sum_n \mathbb{E}_{q_{\phi_d}(z_{nd}|x_{nd})} \log \frac{p_{\theta_d}(x_{nd}, z_{nd})}{q_{\phi_d}(z_{nd}|x_{nd})} \qquad \forall d \in \{1, 2, ..., D\}, \tag{9}$$

Together, the $D$ individual VAEs define a joint distribution over $\mathbf{x}_n$: $\log p_\theta(\mathbf{x}_n) := \log \sum_{\mathbf{z}_n} \prod_d p_{\theta_d}(\mathbf{x}_n|\mathbf{z}_n)p_0(\mathbf{z}_n)$, where $p_0(\mathbf{z}_n)$ is a factorized standard normal distribution. Note that $\mathbb{E}_{q_{\phi_d}(z_{nd}|x_{nd})} \log \frac{p_{\theta_d}(x_{nd}, z_{nd})}{q_{\phi_d}(z_{nd}|x_{nd})}$ in Equation 9 is a lower bound of $\log p_{\theta_d}(x_{nd})$, therefore stage one jointly optimizes a valid lower bound of $\log p_\theta(\mathbf{x}_n)$:

$$\log p_\theta(\mathbf{x}_n) \geq \sum_d \mathbb{E}_{q_{\phi_d}(z_{nd}|x_{nd})} \log \frac{p_{\theta_d}(x_{nd}, z_{nd})}{q_{\phi_d}(z_{nd}|x_{nd})} \tag{10}$$

where $p_{\theta_d}(x_{nd}, z_{nd}) = p_{\theta_d}(x_{nd}|z_{nd})p_0(z_{nd})$.

Then we proceed to the second stage. In this stage, the dependency VAE $p_\psi(\mathbf{z}) = \mathbb{E}_{p(\mathbf{h})} p_\psi(\mathbf{z}|\mathbf{h})$, is trained on the latent representations $\mathbf{z}$ provided by the encoders of the marginal VAEs in the first stage:

$$
\begin{aligned}
\mathbf{x}_n &\sim p_{\text{data}}(\mathbf{x}), \\
z_{nd} &\sim q_{\phi_d}(z_d|x_{nd}), \quad \forall d \in \{1, ..., D\}, \\
(\psi^\star, \lambda^\star) &\propto \arg\max_{(\psi, \lambda)} \sum_n \mathbb{E}_{q_\lambda(\mathbf{h}_n|\mathbf{z}_n, \mathbf{x}_n)} \log \frac{p_\psi(\mathbf{z}_n, \mathbf{h}_n)}{q_\lambda(\mathbf{h}_n|\mathbf{z}_n, \mathbf{x}_n)}.
\end{aligned} \tag{11}
$$

In other words, the second stage of VAEM improves $p_0(\mathbf{z})$ (a factorized standard Gaussian) by $p_\psi(\mathbf{z})$. Since we optimizes the ELBO of $p_\psi(\mathbf{z})$, if we can show that

$$
\mathbb{E}_{q_{\lambda^\star}(\mathbf{h}|\mathbf{z},\mathbf{x})} \log \frac{p_{\psi^\star}(\mathbf{z}, \mathbf{h})}{q_{\lambda^\star}(\mathbf{h}|\mathbf{z}, \mathbf{x})} > \log p_0(\mathbf{z}) \tag{12}
$$

Then we can conclude that the second stage will improve the lower bound given by the first stage (Equation 10). Next, we show that Equation 12 indeed holds. All we need to do is to initialize $p_{\psi_0}(\mathbf{z})$ so that $p_{\psi_0}(\mathbf{z}) = p_0(\mathbf{z})$, and initialize $q_{\lambda_0}(\mathbf{h}|\mathbf{z}, \mathbf{x})$ so that $q_{\lambda_0}(\mathbf{h}|\mathbf{z}, \mathbf{x})$ is the exact posterior of $p_{\psi_0}(\mathbf{h}|\mathbf{z})$.

Note that it is trivial to show that such initialization is possible. One way to do this is to use relu activation functions for hidden layers in the dependency VAE, and then initialize all the weights biases, log variances in the decoder and encoders to be zero. In this way, the decoder of dependency VAE will ignores the latent variable $\mathbf{h}$ and generates factorized standard Gaussians. The encoder with zero initialization will also give factorized Gaussian, which will be identical to the prior $p(\mathbf{h})$. This is exactly the true posterior $p_{\psi_0}(\mathbf{h}|\mathbf{z})$, since the dependency network decoder completely ignores its input: $p_{\psi_0}(\mathbf{z}|\mathbf{h}) = p_{\psi_0}(\mathbf{z})$.

Note that there are many ways to achieve the Equation 12, the above is only one way to do this. The above zero initialization setting is exactly what we have used in our experiments. Finally, we can ensure that by optimizing Equation 11, we always have:

$$
\begin{aligned}
\sum_n \log p_{\psi^\star}(\mathbf{z}_n) &\geq \sum_n \mathbb{E}_{q_{\lambda^\star}(\mathbf{h}_n|\mathbf{z}_n,\mathbf{x}_n)} \log \frac{p_{\psi^\star}(\mathbf{z}_n, \mathbf{h}_n)}{q_{\lambda^\star}(\mathbf{h}_n|\mathbf{z}_n, \mathbf{x}_n)} \\
&> \sum_n \mathbb{E}_{q_{\lambda_0}(\mathbf{h}_n|\mathbf{z}_n,\mathbf{x}_n)} \log \frac{p_{\psi_0}(\mathbf{z}_n, \mathbf{h}_n)}{q_{\lambda_0}(\mathbf{h}_n|\mathbf{z}_n, \mathbf{x}_n)} = \sum_n \log p_{\psi_0}(\mathbf{z}_n) = \sum_n \log p_0(\mathbf{z}_n)
\end{aligned} \tag{13}
$$

Therefore, we finally have:

$$
\begin{aligned}
&\sum_n \log \sum_{\mathbf{z}_n} \prod_d p_{\theta_d}(x_{nd}|z_{nd}) p_{\psi^\star}(\mathbf{z}_n) \\
&\geq \sum_n \mathbb{E}_{\prod_d q_{\phi_d}(z_{nd}|x_{nd})} \sum_d \log \frac{p_{\theta_d}(x_{nd}|z_{nd})}{q_{\phi_d}(z_{nd}|x_{nd})} + \sum_n \mathbb{E}_{\prod_d q_{\phi_d}(z_{nd}|x_{nd})} \mathbb{E}_{q_{\lambda^\star}(\mathbf{h}_n|\mathbf{z}_n,\mathbf{x}_n)} \log \frac{p_{\psi^\star}(\mathbf{z}_n, \mathbf{h}_n)}{q_{\lambda^\star}(\mathbf{h}_n|\mathbf{z}_n, \mathbf{x}_n)} \\
&> \sum_n \sum_d \mathbb{E}_{q_{\phi_d}(z_{nd}|x_{nd})} \log \frac{p_{\theta_d}(x_{nd}, z_{nd})}{q_{\phi_d}(z_{nd}|x_{nd})}
\end{aligned} \tag{14}
$$

Where the second row is the ELBO after the second stage, and the third row is the ELBO after the first stage. The first inequality follows from Jensen's inequality, and the second inequality follows from Equation 13. Therefore, the two stage procedure of VAEM always increases the lower bound of true log-likelihood.

<div align="right">□</div>

## B  Two stage training of VAEM vs joint training of Hierarchical VAE

Based on the analysis of Section A.2, the VAEM training procedure optimizes the following ELBO in a two stage manner:

$$\sum_n \log \sum_{\mathbf{z_n}} \prod_d p_{\theta_d}(x_{nd}|z_{nd})p_\psi(\mathbf{z}_n) \geq$$

$$\sum_n \mathbb{E}_{\prod_d q_{\phi_d}(z_{nd}|x_{nd})} \sum_d \log \frac{p_{\theta_d}(x_{nd}|z_{nd})}{q_{\phi_d}(z_{nd}|x_{nd})} + \sum_n \mathbb{E}_{\prod_d q_{\phi_d}(z_{nd}|x_{nd})} \mathbb{E}_{q_\lambda(\mathbf{h}_n|\mathbf{z}_n,\mathbf{x}_n)} \log \frac{p_\psi(\mathbf{z}_n,\mathbf{h}_n)}{q_\lambda(\mathbf{h}_n|\mathbf{z}_n,\mathbf{x}_n)}$$
(15)

In the first stage, it optimizes Equation 15, but initialize $p_\psi(\mathbf{z}_n)$ to standard gaussian $p_0(\mathbf{z}_n)$ and keep it fixed. In the second stage, VAEM also optimizes Equation 15, but now keeps $p_{\theta_d}(x_{nd}|z_{nd})$ and $q_{\phi_d}(z_{nd}|x_{nd})$ fixed, and optimizes $p_\psi(\mathbf{z}_n,\mathbf{h}_n)$ and $q_\lambda(\mathbf{h}_n|\mathbf{z}_n,\mathbf{x}_n)$.

Note that if we directly optimize Equation 15 jointly instead of the two stage method, then $q_{\phi_d}(z_{nd}|x_{nd})$ will not be regularized by standard Gaussian prior $p_0(\mathbf{z}_n)$. As a result, we will lose the uniformity/homogeneity of the $z_{nd}$ ( to exact, the uniformity of aggregated posterior $\frac{1}{N}\sum_n q_{\phi_d}(z_{nd}|x_{nd})$). On the contrary, in our two stage method, the latent representations $z_d$ provided by the marginal VAEs will have similar properties across dimensions. Each of these variables is encouraged to be close to a standard normal distribution, thanks to the regularization effect from the Gaussian prior $p_0(z_{nd})$. To further demonstrate the effect of such regularization, we compare VAEM versus two-layer VAE (with matching latent dimensions) trained jointly on the data imputation task from Section 5.3. We report the imputation error (will be define in Section E.1) below. As shown in Table 3, the two-layer VAEs trained jointly generally give worse result than VAEM.

Table 3: Data imputation error (averaged per variable)

| Method | VAEM | Two-layer VAE |
|---|---|---|
| Bank | **0.111**±**0.00** | 0.113±0.00 |
| Boston | **0.046**±**0.00** | 0.049±0.00 |
| Avocado | **0.145**±**0.00** | 0.146±0.00 |
| Energy | **0.155**±**0.00** | 0.158±0.00 |
| MIMIC | **0.226**±**0.00** | **0.226**±**0.00** |

## C  Enhancing predictive performance of VAEM: training procedure

In order to enhance the predictive performance of VAEM, the following alternative factorization is proposed:

$$p_\theta(\mathbf{x}_O, \mathbf{x}_\Phi) = \mathbb{E}_{\mathbf{x}_{U\setminus\Phi},\mathbf{h}\sim p_\theta(\mathbf{x}_{U\setminus\Phi},\mathbf{h}|\mathbf{x}_O)} p_\gamma(\mathbf{x}_\Phi|\mathbf{x}_O,\mathbf{x}_{U\setminus\Phi},\mathbf{h})p_\theta(\mathbf{x}_O)$$

For compactness, the notation for input $\mathbf{x}_O$ and $\mathbf{x}_i$ to the all partial inference nets $q_\lambda$ will be omitted. Note that, to train this model, we also need data samples of $\mathbf{x}_\Phi$ during training (however $\mathbf{x}_\Phi$ will not be observed during active learning task). This model is trained using the following procedure:

- Train a partial VAEM on $\mathbf{x}_O$ ($\mathbf{x}_\Phi \cap \mathbf{x}_O = \varnothing$) using the two-stage method described in Section 2.2. Now we have a graphical model induced by the model $p_\theta(\mathbf{x}_O)$.

- Expand the graph by adding the node $\mathbf{x}_\Phi$ to the graph. Now the joint distribution is defined as $p_\theta(\mathbf{x}_O, \mathbf{x}_\Phi) = \mathbb{E}_{\mathbf{x}_{U\setminus\Phi},\mathbf{h}\sim p_\theta(\mathbf{x}_{U\setminus\Phi},\mathbf{h}|\mathbf{x}_O)} p_\gamma(\mathbf{x}_\Phi|\mathbf{x}_O,\mathbf{x}_{U\setminus\Phi},\mathbf{h})p_\theta(\mathbf{x}_O)$. Note that no new parameters need to be introduced for the partial inference net of the dependency network $q_\lambda(\mathbf{h}|\mathbf{z}_O,\mathbf{z}_\Phi)$, since the partial inference net automatically handles inputs with different dimensionalities.

- Define the marginal VAE encoder for $x_\Phi$ as $q_d(z_\Phi|x_{n,\Phi},\phi_\Phi) = \delta(z_\Phi - x_\Phi)$, and the decoder to be $p_d(x_{n,\Phi}|z_d,\theta_\Phi) = \delta(x_\Phi - z_\Phi)$ (i.e., both are identity deterministic mappings).

- The partial inference net parameters of the dependency network can be updated by the following procedure:

$$z_d \sim q_d(z_d|x_{\text{data},d}, \phi_d) \ \ \forall d \in O \cup \Phi, \ \ \mathbf{z}_{O \cup \Phi} = \{z_d | d \in O \cup \Phi\}$$

$$\Delta\lambda \propto \nabla_\lambda \mathbb{E}_{q_\lambda(\mathbf{h}|\mathbf{z}_{O \cup \Phi})} \left[ \log \frac{\prod_{d \in O} p_\psi(z_d|\mathbf{h}) p(\mathbf{h})}{q_\lambda(\mathbf{h}|\mathbf{z}_{O \cup \Phi})} + \mathbb{E}_{\mathbf{x}_{U \setminus \Phi} \sim p_{\theta,\psi}(\mathbf{x}_{U \setminus \Phi}|\mathbf{h})} \log p_\gamma(\mathbf{x}_\Phi|\mathbf{x}_O, \mathbf{x}_{U \setminus \Phi}, \mathbf{h}) \right]$$

- The the parameters for $p_\gamma(\mathbf{x}_\Phi|\mathbf{x}_O, \mathbf{x}_{U \setminus \Phi}, \mathbf{h})$ can be updated by the following procedure:

$$z_d \sim q_d(z_d|x_{,d}, \phi_d) \ \ \forall d \in O, \ \ \mathbf{z}_O = \{z_d | d \in O\}$$
$$\mathbf{h} \sim q_\lambda(\mathbf{h}|\mathbf{z}_O)$$
$$z_s \sim p_\psi(z_s|\mathbf{h}) \ \ \forall s \in U \setminus \Phi, \ \ \mathbf{z}_{U \setminus \Phi} = \{z_s | s \in U \setminus \Phi\}$$
$$x_s \sim p_\theta(x_s|\mathbf{z}_U, \mathbf{z}_O) \ \ \forall s \in U \setminus \Phi, \ \ \mathbf{x}_{U \setminus \Phi} = \{x_s | s \in U \setminus \Phi\}$$
$$\gamma^\star = \arg\max_\gamma \log p_\gamma(\mathbf{x}_\Phi|\mathbf{x}_O, \mathbf{x}_{U \setminus \Phi}, \mathbf{h})$$

# D   Additional Experiment Settings

subsectionDatasets details We use the same collection of mixed type datasets in all tasks:

- Two standard UCI benchmark datasets: Boston housing (13 continuous, 1 categorical) and energy efficiency (6 continuous, 3 categorical) [2];
- Two relatively large real-world dataset: Bank marketing (45211 instances, 11 continuous, 8 categorical, 2 discrete); [21] and Avocado sales prediction (18249 instances, 9 continuous, 5 categorical).
- One real-world medical dataset: Medical Information Mart for Intensive Care (MIMIC III) [10], the largest public medical dataset containing records of 21139 patients (after processing following [7]). We focus on the mortality prediction task based on 17 medical instruments (13 continuous, 4 categorical). Since the dataset is imbalanced (over 80 % of the data has mortality = 0), we balance the dataset by down-sampling the majority class. The time-series observations are averaged to obtain iid data points.

## D.1   Additional model specification

### D.1.1   Baselines: general information

We have used the following baselines in our experiments:

- Heterogeneous-Incomplete VAE (HI-VAE) [22]. We adopt the multi-head structure of HI-VAE and match the dimensionality of latent variables to be the same as our VAEM. HI-VAE is an important baseline, since it is motivated in a similar way as our VAEM, but all marginal VAEs are trained jointly rather as opposed to our two-stage method. We denote this by `VAE-HI`
- VAE: A vanilla VAE equipped with a VampPrior [30]. The number of latent dimensions is the same as in the second stage of VAEM. We denote this by `VAE`.
- VAE with extended latent dimension: Note that the total number of latent variables of VAEM is $D + L$, where $D$ and $L$ are the dimensionalities of the data and the latent space, respectively. This baseline is like the previous one, but with the latent dimension given by $D + L$. We denote this baseline by `VAE-extended`.
- VAE with automatically balanced likelihoods. This baseline tries to automatically equal the scale of each likelihood term of the different variable types in the ELBO by multiplying each likelihood term with an adaptive constant (Appendix D.1). We denote this baseline by `VAE-balanced`.

### D.1.2   Baseline: VAE with balanced likelihoods

This baseline is a naive strategy that tries to automatically balance the scale of the log-likelihood values of different variable types in the ELBO, by adaptively multiplying a constant before likelihood

terms. More specifically, consider the variational lower bound (ELBO) of vanilla VAE:

$$\log p_\theta(\mathbf{x}) \geq \mathbb{E}_{q_\phi(\mathbf{z}|\mathbf{x})} \log \frac{p_\theta(\mathbf{x}, \mathbf{z})}{q_\phi(\mathbf{z}|\mathbf{x})}$$

$$= \sum_{s \in \mathcal{P}} \mathbb{E}_{q_\phi(\mathbf{z}|\mathbf{x})} \log \frac{p_\theta(\mathbf{x}_{s \in \mathcal{P}}, \mathbf{z})}{q_\phi(\mathbf{z}|\mathbf{x})}$$

Where $\mathcal{P}$ is the set of variable types (e.g., continuous, categorical), and $\mathbf{x}_s$ is the set of variables that belong to $s$-th type. In VAE with balanced likelihoods, we weight each likelihood terms by $\{\beta_1, \beta_2, ..., \beta_{|\mathbf{P}|}\}$:

$$\sum_{s \in \mathcal{P}} \beta_s \mathbb{E}_{q_\phi(\mathbf{z}|\mathbf{x})} \log \frac{p_\theta(\mathbf{x}_{s \in \mathcal{P}}, \mathbf{z})}{q_\phi(\mathbf{z}|\mathbf{x})}$$

Where $\sum_s \beta_s = 1$, such that:

$$\beta_s \mathbb{E}_{q_\phi(\mathbf{z}|\mathbf{x})} \log p_\theta(\mathbf{x}_s|\mathbf{z}) = \beta_t \mathbb{E}_{q_\phi(\mathbf{z}|\mathbf{x})} \log p_\theta(\mathbf{x}_t|\mathbf{z}), \;\; \forall s, t \in \mathcal{P}$$

At each epoch of training, a mini-batch/or the entire dataset $\{\mathbf{x}_j\}_{1 \leq j \leq J}$ is selected, and $\beta_s$ are estimated such that:

$$\beta_s \sum_j \mathbb{E}_{q_\phi(\mathbf{z}_j|\mathbf{x}_j)} \log p_\theta(\mathbf{x}_{j,s}|\mathbf{z}_j) = \beta_t \sum_j \mathbb{E}_{q_\phi(\mathbf{z}_j|\mathbf{x}_j)} \log p_\theta(\mathbf{x}_{j,t}|\mathbf{z}_j), \;\; \forall s, t \in \mathcal{P}$$

In our experiments, at each epochs we used full dataset to compute the weights, whenever applicable.

### D.1.3 Likelihood function specification

In this paper, we consider three variable types: continuous, categorical, and discrete. For continuous and categorical variables, we follow the specification of [22]. In other words, to specify the likelihood function of all VAE decoders $p_{\theta_d}(x_{nd}|z_{nd})$ in our paper, we use Gaussian likelihood with constant observational noises $p_{\theta_d}(x_{nd}|z_{nd}) = \mathcal{N}(x_{nd}; \mu(z_{nd}), \sigma^2)$ for continuous data; and for categorical data, we use categorical likelihood with one-hot representation $p_{\theta_d}(x_{nd}|z_{nd}) = \langle \mathbf{l}(z_{nd}), \mathtt{one\text{-}hot}(x_{nd}) \rangle$, where $\mathbf{l}(z_{nd})$ is soft-max output of the decoder.

For discrete variables, we consider two different scenarios: continuous-discrete and ordinal-discrete. Continuous-discrete means that the variable is continuous by its nature, but only discretized values are recorded. For example, the salary (dollars) is a continuous variable, but in practice only discretized values (5000 dollars, 6000 dollars, etc.) are recorded. For this type of variables, we still use Gaussian likelihood, but the decoder output will be rounded to the closest discrete value. On the other hand, ordinal-discrete variables (such as ratings) are the ones with natural orderings, and the distance between each value is not known. For ordinal variables, we use ordinal regression likelihood used in [24].

Note that the above settings are used for all models including VAEM and other baselines.

### D.1.4 Partial inference net with non-continuous input

. In section 3.2, the partial inference net $q_\lambda(\mathbf{h}|\mathbf{z}_O, \mathbf{x}_O)$ is constructed based on the element-wise multiplication operation $\mathbf{s}_O := \{v \times \mathbf{e}_v | v \in \mathbf{z}_O \cup \mathbf{x}_O\}$. How is $v \times \mathbf{e}_v$ defined if $v$ is non-continuous? For categorical and ordinal-discrete variable for example, the operation $v \times \mathbf{e}_v$ is defined as

$$v \times \mathbf{e}_v := vec(\mathtt{one\text{-}hot}(v) \otimes \mathbf{e}_v)$$

Where $\otimes$ is outer-product between vectors, $\mathtt{one\text{-}hot}$ is the one-hot representation of the categorical/ordinal variables, and $vec(\cdot)$ is the vectorization operation of a matrix.

### D.2 Network structure and hyper parameter settings

**Network structures** All models (except for the marginal VAEs of VAEM and the decoder of HI-VAE) share the same network structures with 20 dimensional diagonal Gaussian latent variables: the generator (decoder) is a 20-50-100 fully connected neural network with ReLU activation functions on hidden units (where $D$ is the data dimension). Note that we use sigmoid activation function for

output layer, to reflect our data preprocessing (all data are normalized to between 0 and 1). One exception is the output layer of dependency network of VAEM, where we did not add any activation functions since the scales of the latent variables $z_d$ from marginal VAEs are unknown. The encoders share the same structure of $D$-500-200-40 that maps the observed data into distributional parameters of the latent space. Additionally, we use a $K = 100$ dimensional feature mapping parameterized by a single layer neural network, and $M = 10$ dimensional feature embedding for each variable. We choose the permutation invariant operator $g$ to be the summation operator. The discriminator described in section 3.4 is a neural network with two layers, each of which has 100 hidden units.

For marginal VAEs of our VAEM, we use 1-dimensional latent variable for each variable.The decoder of marginal VAEs is a 1-50-V single layer neural network, and the encoder network structure is V-50-2, where $V$ is the dimension of the corresponding variable, which is defined to be 1 if the variable is continuous. Otherwise, $V$ is the dimension of the one-hot representation. The same structure is used for the multi-head decoder structure for HI-VAE baseline.

**Hyperparameters**    To train our models, we apply Adam optimization [14] with learning rate of 0.001 and a batch size of 100. When the training set is fully observed, We manually generate partially observed version of it by adding artificially missingness at random in the training dataset during training. This will help the model to learn to generate conditional data given observations. We first draw a missing rate parameter from a uniform distribution $\mathcal{U}(0, 1)$ and randomly choose variables as unobserved. This step is repeated at each iteration. We train our models for 3000 full epochs, except for Bank dataset where we used 5000 epochs. For continuous variables, the constant observational noise variance level for Gaussian likelihood functions of decoders are set to be 0.02 (except for MIMIC dataset where we have used 0.3). During evaluation, we use importance sampling with 10K samples to estimate the log-likelihoods for conditional data generation.

**Sequential active information acquisition**    For SAIA tasks, we use 10 monte-carlo samples from VAE models to estimate reward functions. Since the focus of this paper is comparing the performance of different generative models on heterogeneous mixed type data, we use the SING strategy [18] for SAIA, which uses the objective function as in Equation (7) to find the optimal ordering, by averaging over all the currently observed test data.

### D.3   Additional experiment pipeline setup

In Section 5.2, during training, the range of all variables is scaled to be between 0 and 1. This transformation is removed when making predictions on the target variables.

In Section 5.3, to train these partial models on data with missing values, we randomly sample 90% of the dataset to be the training set, and assume that a random fraction (uniformly sampled between 0% and 99%) of feature values are missing on each epoch during training. Then, during test time, we assume that 50% of the test set is observed, and use generative models to infer the unobserved data.

# E    Additional experimental results

## E.1    Imputation errors of conditional data generation experiment

Here we also provide results that uses imputation errors to evaluate model performance in Section 5.3.

Note that one issue with imputation error is: since now we have mixed type data, the errors of different variables are not directly comparable. Therefore, one often need to introduce a coefficient to weight the error of different types of variables. The ranking of imputation performance will be highly dependent on the choice of such coefficient.

Here, we set the weighting coefficients to be 1, and calculate the imputation error based on RMSE. For continuous variables, the RMSE/SE is defined as usual; for categorical variables, the RMSE/SE will be calculated based on their one-hot encodings. Then, we take the average of errors across all variables as our final metric. The calculation are specified as follows:

$$\frac{1}{D}\sqrt{\sum_{1 \leq d \leq D} \sum_{1 \leq n_d \leq N_d} \frac{SE(x_{n_d,d} - \hat{x}_{n_d,d})}{N_d}}$$

Where $D$ is the number of features, $N_d$ is the number of unobserved slots to be imputed for $d$th variable. $SE$ stands for squared errors. The results are summarized in Table 4. We can see that the results are consistent with our NLL evaluations in Table 2 from our main text.

Table 4: Data imputation error on Bank dataset (averaged per variable)

| Method | Ours | VAE | VAE-balanced | VAE-extended | VAE-HI |
|---|---|---|---|---|---|
| Bank | **0.111**±**0.00** | 0.116±0.00 | 0.117±0.00 | 0.116±0.00 | 0.113±0.00 |
| Boston | **0.046**±**0.00** | 0.048±0.00 | 0.098±0.00 | **0.046**±**0.00** | 0.054±0.00 |
| Avocado | **0.145**±**0.00** | 0.145±0.00 | 0.179±0.00 | 0.145±0.00 | 0.146±0.00 |
| Energy | **0.155**±**0.00** | 0.176±0.00 | 0.187±0.00 | 0.184±0.00 | 0.176±0.00 |
| MIMIC | **0.226**±**0.00** | 0.228±0.00 | 0.230±0.00 | 0.229±0.00 | **0.226**±**0.00** |
| Avg. Rank | **1.00**±**0.00** | 2.40±0.40 | 5.00±0.00 | 2.60±0.67 | 2.60±0.60 |

## E.2    Approximation errors of marginal VAEs

One of the main differences between our VAEM and vanilla VAEs is that we introduce one additional marginal VAE per data dimension. We evaluated the posterior approximation quality in these marginal VAEs in the Avocado dataset. The table below shows that, in each marginal VAE, the ELBO and log-likelihood are very similar:

Table 5: ELBO vs log likelihood of marginal VAEs

| Variables | 1 | 2 | 3 | 4 | 5 | 6 | 7 | 8 |
|---|---|---|---|---|---|---|---|---|
| ELBO | -1.30 | -4.21 | -2.43 | -3.49 | 1.97 | 2.11 | 2.12 | 2.17 |
| LLH | -1.30 | -4.17 | -2.42 | -3.48 | 2.00 | 2.14 | 2.13 | 2.19 |

Since the gap between ELBO and LL is the KL divergence, we conclude that the posterior approximation quality is very high in this case.

In addition, our results (e.g. Figure 1 in the paper) show that our method approximates the marginal distributions of the data better than vanilla VAE.

# F   Additional Plots on Bank dataset

Figure 5: pair plots of all variables from the real Bank dataset. Diagonal plots show marginal histograms for each variable. The upper-triangular part shows sample scatter plots for each variable pair. The lower-triangular part shows heat maps identifying regions of high-probability density for each variable pair. For visualization, categorical variables are mapped to a grid of evenly spaced points in the interval $[0, 1]$.

Figure 6: pair plots of all variables generated by VAEM. Diagonal plots show marginal histograms for each variable. The upper-triangular part shows sample scatter plots for each variable pair. The lower-triangular part shows heat maps identifying regions of high-probability density for each variable pair. For visualization, categorical variables are mapped to a grid of evenly spaced points in the interval $[0, 1]$.

Figure 7: pair plots of all variables generated by VAE-balanced. Diagonal plots show marginal histograms for each variable. The upper-triangular part shows sample scatter plots for each variable pair. The lower-triangular part shows heat maps identifying regions of high-probability density for each variable pair. For visualization, categorical variables are mapped to a grid of evenly spaced points in the interval $[0, 1]$.

Figure 8: pair plots of all variables generated by HI-VAE. Diagonal plots show marginal histograms for each variable. The upper-triangular part shows sample scatter plots for each variable pair. The lower-triangular part shows heat maps identifying regions of high-probability density for each variable pair. For visualization, categorical variables are mapped to a grid of evenly spaced points in the interval $[0, 1]$.

Figure 9: pair plots of all variables generated by VAE-extended. Diagonal plots show marginal histograms for each variable. The upper-triangular part shows sample scatter plots for each variable pair. The lower-triangular part shows heat maps identifying regions of high-probability density for each variable pair. For visualization, categorical variables are mapped to a grid of evenly spaced points in the interval $[0, 1]$.