[Reviews · NeurIPS 2020]

Review 1

Summary and Contributions: Summary: Variational auto-encoders (VAEs) are a powerful approach to probabilistic modeling. However they are not directly amenable to heterogenous data. This paper proposes to adapt VAEs to these types of data. The idea is to use a two-stage procedure. First fit a VAE to each dimension of the data. Then capture the dependencies by fitting a new VAE to the individual latents. Interestingly, this two-step procedure optimizes a lower bound to the log marginal likelihood of the data. The proposed method is tested on image generation on five different datasets. The method is then extended to handle missing data and this extension is tested on conditional data generation and sequential information acquisition problems. The results show the proposed method avoids the problems of VAEs in handling heterogenous data. States Contributions: 1) propose a new family of VAEs for heterogenous data. 2) extends the family proposed in 1) to handle missing data imputation with application to conditional data generation and sequential information acquisition. ----- After rebuttal: I really enjoyed reading the paper. I am keeping my decision to accept this paper. Good work.

Strengths: Very well motivated paper. Very well written. Nice applications for the empirical study.

Weaknesses: Needing to fit D different VAEs might seem suboptimal, especially when the data is very high-dimensional. But this should not be a big problem since the different VAEs are fit to one-dimensional variables.

Correctness: I didn't see any incorrect statements or flaws in the methodology.

Clarity: The paper is very well written, very well motivated, and very easy to follow and understand. It was a really great read.

Relation to Prior Work: Very nice related work section.

Reproducibility: Yes

Additional Feedback: Questions & Remarks: 1--What happens when D is very large (high-dimensional data)? You are going to train D different "marginal VAEs" or a VAE for each modality? That seems like a lot. Have you tried fitting simpler models to each dimension instead and using the dependency network on the resulting latents? 2--It looks to me as though the way you model the latent space relates to copulas. With copula modeling you model the marginals separately and then capture all the correlation in a covariance matrix. Is it possible to make this connection if there is indeed one? 3--The reward function R_I defined right after line 161 is the mutual information between x_i and the target, conditional on the observations. So maximizing this reward is equivalent to selecting the next point that maximizes its mutual information with the target. That's the intuition I believe. 4--Suggestion: maybe you can call your method mixVAE? I am not sure whether that name is taken already but it is catchier than VAEM.


Review 2

Summary and Contributions: - This paper proposed a new VAE-based approach (dubbed “VAEM”) to handle datasets that contain variables of different types (Categorical & Continuous). - The authors show that why training VAEs naively (without any adjustment to the likelihood model) can results in bad data generation. - The authors then perform a set of experiments to show that VAEM can successfully address this issue to some extent.

Strengths: I find the problem stated in this paper to be a novel that has not been paid attention to much. Most of the work on VAEs and other deep generative models are on images where the community fights to improve the generative model by making the architecture or the variational distributions more complex. However, as the authors point out, many real-world datasets contain variables with different types, so I think it is valuable to design models to handle these cases. The extensions to the VAEM model such as the new reward estimator and training another network with eq.8 seems to involve a significant amount of work.

Weaknesses: Could the authors clarify something? According to Figure 2.b VAEM trains a VAE for EVERY dimension of x. While the authors are correct that in most real-world datasets, data contains variables of different types, I would argue that it is also the case that most real-world datasets are high dimensional. The dataset chosen for the experiments contain no more than 20 variables. Training that many VAEs is certainly not feasible. Is it the case these models are trained in parallel? Or is it the case that you group all variables of the same type together and train a single VAE on those (for the 1st stage)?

Correctness: Yes the objective makes sense.

Clarity: This paper reads well overall. Maybe you can just add a normal legend for Figure 4? It makes the finger hard to read when the legend is broken to be at the top of all 6 plots. Typo: “Optimization objective” on page 4

Relation to Prior Work: I’m not very familiar with this line work. However two set of models that seems relevant to me are: 1) VAE with categorical variables [1,2,3,4]: these models also deal with continuous and categorical variables so this is done in latent space which is different than the proposed model 2) Multimodal representation learning [4,5]: The data in these settings also come in deterrent forms e.g. (image, caption) or (video, audio). Similar to this paper, these models also have a shared network and a network for different modalities. Could the authors comment on these? [1] Dupont, Emilien. "Learning disentangled joint continuous and discrete representations." Advances in Neural Information Processing Systems. 2018. [2] Esmaeili, Babak, et al. "Structured disentangled representations." The 22nd International Conference on Artificial Intelligence and Statistics. 2019. [3] Kingma, Durk P., et al. "Semi-supervised learning with deep generative models." Advances in neural information processing systems. 2014. [4] Siddharth, Narayanaswamy, et al. "Learning disentangled representations with semi-supervised deep generative models." Advances in Neural Information Processing Systems. 2017 [5] Shi, Yuge, et al. "Variational mixture-of-experts autoencoders for multi-modal deep generative models." Advances in Neural Information Processing Systems. 2019. [6] Wu, Mike, and Noah Goodman. "Multimodal generative models for scalable weakly-supervised learning." Advances in Neural Information Processing Systems. 2018.

Reproducibility: Yes

Additional Feedback: ======== Update ======== I thank the authors for their response. My concern regarding dimensionality has been addressed. I'm still happy to recommend this paper for acceptance.


Review 3

Summary and Contributions: This paper proposes VAEM, a two-level hierarchical VAE designed to handle heterogeneous data (e.g. mix of categorical and continuous features). Specifically, the first layer of VAEM consists of D univariate VAEs (which the paper calls "marginal VAEs") each modeling a single dimension of the D-dimensional input independently from the rest. The second layer (called the "dependency network") is another VAE that further models the distribution of the D latent variables from the first level. The paper also discusses the application of VAEM on sequential active information acquisition (SAIA), the task of determining which variable should be observed given a partial observation to maximize information gain measured by a predefined "information reward" function. Experimental results on density estimation and SAIA on heterogeneous data are provided. --------- Post-rebuttal: I have reviewed the author feedback, and many of my concerns were addressed. I'm increasing the score to 5.

Strengths: Motivation and Significance - The paper is well-motivated, as it tackles the problem of training a VAE on heterogeneous mix of variables, which many real-world applications involve. Much of the existing work on VAEs (and more broadly generative modeling) focuses on homogeneous data, and there is certainly a need for methods that can model heterogeneous data.

Weaknesses: Formulation & Novelty - Architecture: I am a bit concerned with the novelty of the proposed model. The only difference between a vanilla hierarchical VAE and VAEM is in the first layer, which contains individual "marginal VAEs" for each of dimensions. This approach of having a separate network for each dimension is also similar to the architecture proposed in HI-VAE [1], as acknowledged in Section 4. - Training procedure: Since the training of the two layers happens in distinct stages, it brings the question of whether this training procedure would lead to suboptimal decoder and encoder upon completion compared to joint training. Appendix A.2 has a brief discussion on this and argues that jointly training both layers may lose the "uniformity" of each latent variable z_{nd}, but I'm not sure what "uniform properties" the authors are referring to. Empirical evaluation - Datasets: As the highlight of the paper is VAEM's ability to model heterogeneous data, it'd be very helpful to mention what kind of heterogeneity is present in the datasets used for empirical evaluation. - Likelihood scaling: In the introduction, the paper claims that naively training a VAE on heterogeneous data can be difficult because "the contribution that each likelihood makes to the training objective can be very different, leading to challenging optimization problems in which some data dimensions may be poorly-modeled in favor of others". This naturally brings up the question of whether careful tuning of the scaling coefficient for the likelihood function of each dimension could ease the aforementioned optimization difficulties. The "VAE-adaptive" baseline seems to be a data-dependent attempt at this, but I'm not convinced that a single minibatch is sufficient for computing the coefficients for each data type (as described in Appendix C.1.2). In particular, it'd be interesting to see if VAEM would outperform a (possibly hierarchical) VAE with more carefully tuned scaling factors for each dimension to rule out the possibility that the poor performance of vanilla VAE baselines is simply due to hyperparameter tuning. - Ablation study: To empirically verify that the proposed approach indeed improves performance of other VAE models, it would have been nice to see an ablation test on a few different architectures. For example, a more powerful VAE model (e.g. Ladder VAE) could be used to model the data with or without the first layer containing marginal VAEs. - Choice of evaluation task: It's unclear why SAIA was chosen as an evaluation metric when the focus is on handling the heterogeneity in data. Could authors elaborate further on the motivation behind why SAIA was used? [1] Nazabal, Alfredo, et al. "Handling incomplete heterogeneous data using vaes." Pattern Recognition (2020): 107501.

Correctness: Experiment setup - Baseline architecture: The comparison made between VAEM and vanilla VAE does not seem fair, given that VAEM has two layers of hierarchical latent variables whereas both VAE and VAE-extended have a single layer of latent variable. It would be nice to see how VAEM compares to a two-layer hierarchical VAE with matching latent dimensions.

Clarity: The overall writing quality of the paper is good, although there are a few typos throughout the paper. These typos very minor, however, and do not obfuscate the clarity of the exposition.

Relation to Prior Work: The paper properly discusses how it differs from existing work.

Reproducibility: Yes

Additional Feedback: - In Table 1, why are some negative log-likelihood values negative? For discrete variables (i.e. categorical and ordinal), the corresponding marginal probability mass should be at most 1. So the only way for the NLL to be negative would be if the model assigned a very high density on some values for a continuous variable (which is definitely possible when the model overfits, but also possible even when it doesn't). This may not be an issue, just wanted to double check that these numbers made sense. - For the VampPrior used in the models. how were the pseudo-inputs u_k chosen? Specifically, the last sentence of Section 2.2 (line 102) states that they are a subset of data points. But how were those data points chosen?


Review 4

Summary and Contributions: The authors propose a deep generative model to handle heterogeneous data (as in different likelihood models) that uses a two-stage procedure: first train a series of standard VAEs for each univariate marginals in parallel, then combine them by training a globall VAE on each. In such a way the first stages just perform Gaussianization for marginals, delivering an homogeneous space for the global, second VAE to perform (approximate) density estimation.

Strengths: The major strength of the paper is providing evidence that the two-stage approach in VAEs can help alleviate the misproportioned contributions of the different likelihood models.

Weaknesses: One weak point is the presentation: the first stage idea is not really dependent on VAEs and can be realized in several ways, for instance the one in [1] which provides a more principled way to deal with heterogeneous likelihood models. In fact, the current first/local stage is just a mean for Gaussianization but it is agnostic of the statistical data type of the input distribution and assumes inputs are only Reals with infinite support or Categoricals. [1] - Valera, Isabel, and Zoubin Ghahramani. "Automatic discovery of the statistical types of variables in a dataset." International Conference on Machine Learning. 2017.

Correctness: The claims and derivations I checked are correct. More discussion about related approaches should be carried out. On the motivational side, it is not clear why one should adopt a VAE, very limited in performing inference (MAP, marginals, etc) w.r.t. other tractable alternatives. Indeed, the proposed 2 staged VAE resuses the same heuristics of the Partial VAE for handling missing values, which provide no guarantees on inference. I would say the advantage of using a VAE could lie in re-using the latent embeddings but this does not seem the case for the paper. Concerning experiments, see more comments below.

Clarity: The paper is well-written, clear and easy to follow.

Relation to Prior Work: The paper correctly points to Gaussianization for the role of the first stage VAEs. I believe the overall presentation would benefit from a deeper discussion on the topic, showing how one does not necessarily need a VAE for trasforming single variables into Gaussians. A Flow could do the job, or even the transformations in [1]. Furthermore, if one used deterministic transformations, only density estimation in the joint latent space would be required [2, 3] (or [4] if one applies only the two stage procedure fashion) [2] Kumar, Abhishek, Ben Poole, and Kevin Murphy. "Regularized autoencoders via relaxed injective probability flow." arXiv preprint arXiv:2002.08927 (2020). [3] Brehmer, Johann, and Kyle Cranmer. "Flows for simultaneous manifold learning and density estimation." arXiv preprint arXiv:2003.13913 (2020). [4] Ghosh, Partha, et al. "From variational to deterministic autoencoders." arXiv preprint arXiv:1903.12436 (2019).

Reproducibility: Yes

Additional Feedback: The assumption of categorical data for discrete is a limitation. What happens when the discrete feature comes from a distribution with larger (even infinite, e.g. Poisson) support from the one observed during training? Are integer/count data in the UCI datasets considered to be categorical or real? Why not considering a variant where the two stages are trained end-to-end? I can imagine that this can be highly instable, but would be a useful baseline in the experiments. Likelihoods can be highly misleading w.r.t. sample quality when dealing with continuous data, even more in the heterogeneous setting. What about checking the quality of generated samples in terms of a statistical test? Or even missing values imputation as in [1]. This is partially done in the active learning scenario, but the effect of performing MAP inference correctly is harder to disentangle from the effect of the online learning. --------- UPDATE I thank the authors for their answers, I believe the paper is worth acceptance and the additional work in presentation can be done in the camera-ready

[Author Response · NeurIPS 2020]

We thank the reviewers (R) for their insightful comments. We acknowledge that the reviewers highlighted the importance
of our motivation (R1, R2, R3,R4), the significance and simplicity of our methodology (R1,R2,R4 ), and the soundness
of empirical evaluations (R1,R2,R4). We address the concerns of each reviewer below.

R1 & R2: Do marginal VAEs scale to high dim data? Yes. The marginal VAE training is highly scalable to high dim
data since the marginal VAEs can be trained in parallel very easily, using simple vectorization tricks. This is how we
implemented our experiments. Moreover, the network size in each marginal VAE is very small as it only need to learn
one dimensional marginal distribution for each feature. We will open-source our code upon acceptance.

R1 & R3: Does two stage training introduce suboptimiality? The suboptimality, if any, comes from the error induced by
marginal VAEs. As pointed out by R1, this should not be a big problem since they are fit to one-dimensional variables.
In appendix D.2, we have evaluated the approximation quality of each marginal VAEs, which is indeed very high. *Also,*
*we have introduced a new baseline where the model is trained jointly. This will be presented later in this rebuttal*

R3 : Is VAEM novel in comparison with HI-VAE? And what does "uniformity" mean? Indeed, VAEM highly relates to
HI-VAE *which is one of our baselines (dubbed as VAE-HI)* in our experiments. In all the experiments, we have shown
that VAEM has a very significant improvement over HI-VAE, confirming the novelty of our contribution. The reviewer
may have missed this baseline due its naming, which we will change from VAE-HI to HI-VAE and clarify accordingly.

To understand the novelty of VAEM, we would like to point out that: **1**, in HI-VAE, the first layer latent representation
$z_{nd}$ are *deterministic*, while in VAEM they are stochastic. **2**, unlike HI-VAE (trained end-to-end), VAEM is trained
in two-stage. Therefore, the *marginal statistics and inter-variable dependencies are separated.* Meanwhile, it's now
possible to introduce prior terms $p(z)$ (Appendix A.2). Thanks to $p(z)$, the marginal distribution for $z_{nd}$ is enforced to
be standard Gaussian, so that the dependency network only has to model random variables that are of the same statistical
type and with homogeneous marginal distributions. This *"Gaussianization" (acknowledged by R4, also referred as*
*"uniformity" in our paper) does not happen with the HI-VAE (nor other more general hierarchical VAE methods).*

R3 & R4: Comparison with two-layer VAE baseline? We acknowledge this suggestion. For completeness, we ran two-
latent-layer VAE (trained jointly, with matching latent dimensions) as baseline on data generation tasks. Other training
hyperparameters are consistent with other baselines. The nllh performance (Bank: 1.678$\pm$.05, Boston: -0.629$\pm$.01,
MIMIC: -0.394$\pm$.00, Avocado: -0.137$\pm$.00, Energy: -1.46$\pm$.01) is generally worse than our method.

R3: Missing description of the heterogeneity in the dataset? We have indeed presented this information in Appendix C
(mentioned in Section 5). All sources of heterogeneity are presented in the data-sets used. You can also see it from the
ground truth data distribution in Appendix E. We will add further information regarding each feature in each dataset.

R3: In VAE-adaptive baseline, a single minibatch is not sufficient to compute the scaling factor. In our VAE-adaptive
baseline, the scaling factors are indeed fine-tuned *using the entire dataset* as suggested. We mentioned the "mini-batch"
approach in Appendix C.1.2. only because it is more general and scalable. We will clarify this. Also, we have tried
to fine-tune each scaling factor manually, which yields similar results, and VAEM still outperforms this baseline
significantly. We did not include this result as it is similar to the VAE-adaptive.

R3: Comparison to more complicated models such as Ladder VAE? Our two-stage VAEM approach is in principle
compatible with any VAE decoders and could also be applied to Ladder VAEs. Other advances in VAE can be applied
to VAEM in the same way as in VAE. To further address the reviewer's concern, we would like to point out that our
HI-VAE baseline has a similar structure/parameter numbers/model complexity compared with VAEM. Also, HI-VAE is
trained end-to-end. Hence, HI-VAE already serves as a baseline for ablation study in this case.

R3 : Why is SAIA used? Besides its high application impact, SAIA is highly relevant since it quantifies how well the
model fits the data and how good the inference is. SAIA can be treated as an extension of imputation tasks, since it
assesses the overall imputation performance of the model without specifying a certain ratio of missing data at test time.
*In a heterogeneous data setting*, if a model cannot handle heterogeneity well, it might favor certain types of features,
resulting in poor performance in the SAIA task.

R3 :Is negative NLLHs a sign of overfitting? Not necessarily. Negative NLLHs are perfectly possible when there are
many continuous variables with highly peaked densities. This is indeed the case in our datasets (Appendix C and E).

R4: relationship to (Valera et al., 2017) and necessity of VAE in the first stage. We have discussed the mentioned work
(Valera et al., 2017) in our paper. It is orthogonal to our work since it addresses the problem of automatic type discovery
in a traditional LVM setting. In our scenario (all types are known), VAEs are particularly useful, since: **1**, VAEs are
very efficient and scalable in practice, and **2**, we need a probabilistic model for down-stream tasks such as SAIA, since
this will enable efficient quantification of information gains.

R4: Likelihood is not sufficient. Why not try missing value imputation tasks? We have included imputation tasks in Sec.
5.3, and provided imputation error metric in Appendix D.

[Meta-Review · NeurIPS 2020]

The paper proposes modelling vectors with dimensions having different types (real-valued and categorical) using a two-stage VAE approach. First, a VAE with a 1D latent is trained once for each input dimension to standardize the data. Then a "dependency" VAE is trained on top of the resulting latents to capture the dependence between them. Pros: -The approach is interesting and novel -The idea is simple and seems effective, so might be widely adopted -The paper is well written -VAEM outperforms sensible baselines at generative modelling and a sequential information acquisition task Cons: -It is not explained why the two-stage training approach is a good idea. The fact that joint training tends to perform less well than two-stage training, as reported in the rebuttal, is an important observation that should be discussed and, ideally, explained in the paper. -The UCI datasets used for the evaluation are very small -The paper does not explore whether the benefits come from better inference, better generation, or both